# WorldTravel: A Realistic Multimodal Travel-Planning Benchmark with Tightly Coupled Constraints

Zexuan Wang[*1]  Chenghao Yang[*1]  Que Yingqi[1]  Zhoufutu Wen[1]  Zaiyuan Wang[2]  Jiashuo Liu[1]  Zhixin Yao[1]
Zhenzhu Yang[3]  Huaqing Yuan[3]  Yiwen Wang[3]  Zhengxuan Jiang[3]  Shengjie Fang[3]  Zhenhe Wu[3]
Zhaohui Wang[3]  JinCheng Ren[3]  Yuzhen Li[1]  Yang Yang[1]  Jiaheng Liu[3]  Jian Yang[3]  Ge Zhang[1]
Wenhao Huang[1]

## Abstract

Real-world autonomous planning requires coordinating tightly coupled constraints where a single decision dictates the feasibility of all subsequent actions. However, existing benchmarks predominantly feature loosely coupled constraints solvable through local greedy decisions and rely on idealized data, failing to capture constraint acquisition from realistic web interfaces. We introduce **WorldTravel**, a benchmark comprising 150 real-world travel scenarios across 5 cities, requiring agents to satisfy an average of 15+ interdependent temporal and logical constraints. To evaluate realistic deployment settings, we further develop **WorldTravel-Webscape**, a multi-modal environment with over 2,000 rendered webpages that preserve layout-dependent and information-dense travel interfaces, requiring agents to recover executable constraints from rendered web interfaces. Evaluating 10 frontier models reveals a severe performance collapse: GPT-5.2 achieves only 28.0% feasibility in text-only settings, dropping to 3.4% in multi-modal environments. We observe substantial degradation in planning feasibility when agents must recover executable constraints from rendered webpages, alongside a Planning Horizon threshold at approximately 10 constraints where reasoning reliability collapses. These findings suggest that realistic constraint acquisition and long-horizon planning remain complementary bottlenecks for current agents.

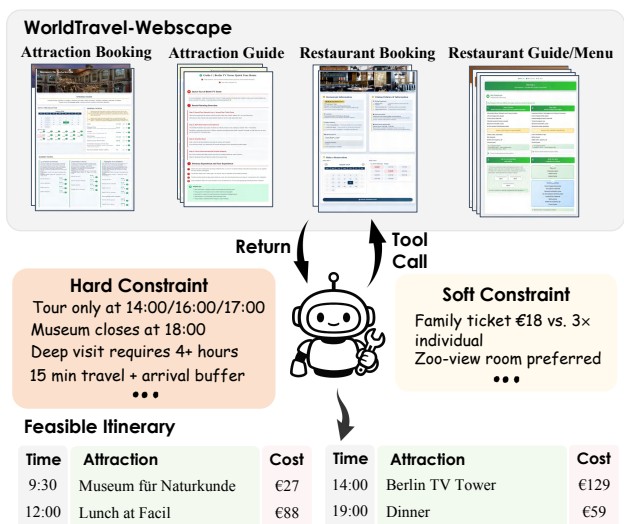

*Figure 1.* **Overview of WorldTravel.** Agents interact with **WorldTravel-Webscape** through tool calls to retrieve constraint parameters from rendered webpages (e.g., attraction booking, restaurant guides). The webpages mirror real booking platforms with realistic UI layouts and interaction patterns (see Appendix K for specifications). The agent must satisfy hard constraints (timed-entry slots, operating hours, dwell times, travel buffers) and soft constraints (cost optimization, lodging preferences) to produce a feasible itinerary.

## 1. Introduction

Large Language Models (LLMs) are rapidly transcending their origins as conversational assistants, evolving into autonomous agents that navigate the intricacies of our daily lives (Wang et al., 2024; Li et al., 2025; Yao et al., 2023b). As their functional boundaries expand, we are increasingly delegating tasks that are fundamentally Operations Research (OR) problems in disguise: from decomposing quarterly sales targets into actionable milestones to orchestrating complex, multi-city travel itineraries. In these roles, the model is no longer just processing text; it is solving a high-dimensional puzzle of interdependent decisions where every action must align with rigid temporal and logical constraints.

---

[*]Equal contribution  [1]ByteDance Seed, China [2]Humanlaya, China [3]M-A-P, China. Correspondence to: Zhoufutu Wen <liniuniu@bytedance.com>, Wenhao Huang <huang.wenhao@bytedance.com>.

*Proceedings of the 43rd International Conference on Machine Learning*, Seoul, South Korea. PMLR 306, 2026. Copyright 2026 by the author(s).

To make this coupling concrete, consider a scenario in one of the world's most logistically demanding cities:

> *"A 10:00 Louvre entry and a 13:00 gallery closure leave only a narrow lunch window before a hard 15:20 Eurostar cutoff. A small delay or a sold-out reservation can cascade into an infeasible itinerary."*

This brittleness, where a single violation can invalidate the entire plan, is precisely what autonomous agents must handle in real deployments. Yet despite the high value of such applications, current evaluation frameworks provide only a partial view of these skills. Existing benchmarks, such as *TravelPlanner* (Xie et al., 2024) and *ChinaTravel* (Shao et al., 2024), often simplify these challenges into independent, "greedy-solvable" subproblems where a local failure does not necessarily compromise global feasibility. Furthermore, most existing benchmarks operate in "clean" settings where constraint information is already normalized into structured formats (e.g., JSON). In realistic deployments, the same constraints are rarely explicit or centralized; they must be recovered from rendered webpages, where key constraints are expressed through visual signals such as sold-out status, tiered pricing, and slot availability. A natural question, then, is *whether current agents can handle the tightly coupled constraints present in real-world web interfaces*.

To rigorously evaluate end-to-end itinerary planning under realistic execution constraints, we introduce **WorldTravel**, a benchmark comprising a task suite, evaluation protocol, and expert-designed rubrics for itinerary planning. Together with **WorldTravel**, we release **WorldTravel-Webscape**, a controlled interaction environment integrating more than 2,000 webpages that mirror real booking platforms. **WorldTravel** contains 150 real-world tasks distributed across five major European cities, while **WorldTravel-Webscape** spans 36 attractions, 25 restaurants, and 26 hotels, enabling agents to retrieve constraint parameters via tool calls during execution. By preserving tightly coupled temporal dependencies and fragmented information under a deterministic notion of feasibility, **WorldTravel** enables reliable evaluation of globally constrained planning across different agent architectures, while task-specific rubrics support fine-grained diagnosis of intermediate decisions and failure modes.

Our evaluation of 10 frontier models reveals a substantial visual-to-plan gap. In a text-only setting where parameters are pre-extracted, the best-performing model, GPT-5.2, achieves a feasibility rate of only 28.0%. When required to extract parameters from rendered webpages in our multimodal setting, performance drops to 3.4%, suggesting that acquiring planning constraints from realistic web interfaces remains challenging for current models. At the same time, even when all constraint parameters are provided explicitly (Gold-$\theta$), planning under tightly coupled constraints remains difficult, with the best model achieving only 26.67% feasibility. Together, these findings suggest that visual constraint acquisition and long-horizon planning pose two distinct challenges for current agents. Through ablation studies, we identify a "Planning Horizon" effect: model performance degrades consistently as the number of constraints increases, with a clear inflection point at approximately 10 constraints.

Our contributions are as follows:

- We introduce **WorldTravel**, a benchmark for end-to-end itinerary planning, comprising 150 real-world scenarios across five European cities with tightly coupled temporal dependencies and globally constrained feasibility.
- We design **WorldTravel-Webscape**, a controlled multimodal interaction environment where agents retrieve constraint parameters from rendered webpages under fragmented real-world settings. Together with task-specific expert-designed rubrics in **WorldTravel**, it supports reliable and deterministic evaluation of globally constrained planning.
- Through experiments and ablations, we identify two major challenges for current agents: a substantial **visual-to-plan gap** when planning moves from text-accessible parameters to rendered webpages, and a **Planning Horizon** effect in which performance deteriorates substantially as constraint complexity increases.

**Conflict of Interest Disclosure** Zexuan Wang and Zhoufutu Wen are employed by ByteDance, which leads the development of Doubao, one of the models evaluated in this paper.

## 2. Related Work

**LLM-Based Agents.** Large language model-based agents have demonstrated remarkable capabilities in executing multi-step tasks (Wang et al., 2024), showing potential across domains such as education (Zhang et al., 2025), healthcare (Zhang et al., 2023), and long-term personalized dialogue agents (Li et al., 2025). To enhance their planning ability, some works decompose complex tasks into subtasks (Shen et al., 2023; Wang et al., 2023), while others focus on step-by-step reasoning strategies for dynamic planning (Wei et al., 2022; Besta et al., 2024; Yao et al., 2023a;b). Long-term memory mechanisms (Zhong et al., 2024; Li et al., 2025) and external tool use (Schick et al., 2023; Qin et al., 2024) further extend agent capabilities by enabling access to historical knowledge and external resources. Despite promising progress, the effectiveness of these approaches under tightly coupled, constraint-intensive planning scenarios remains uncertain.

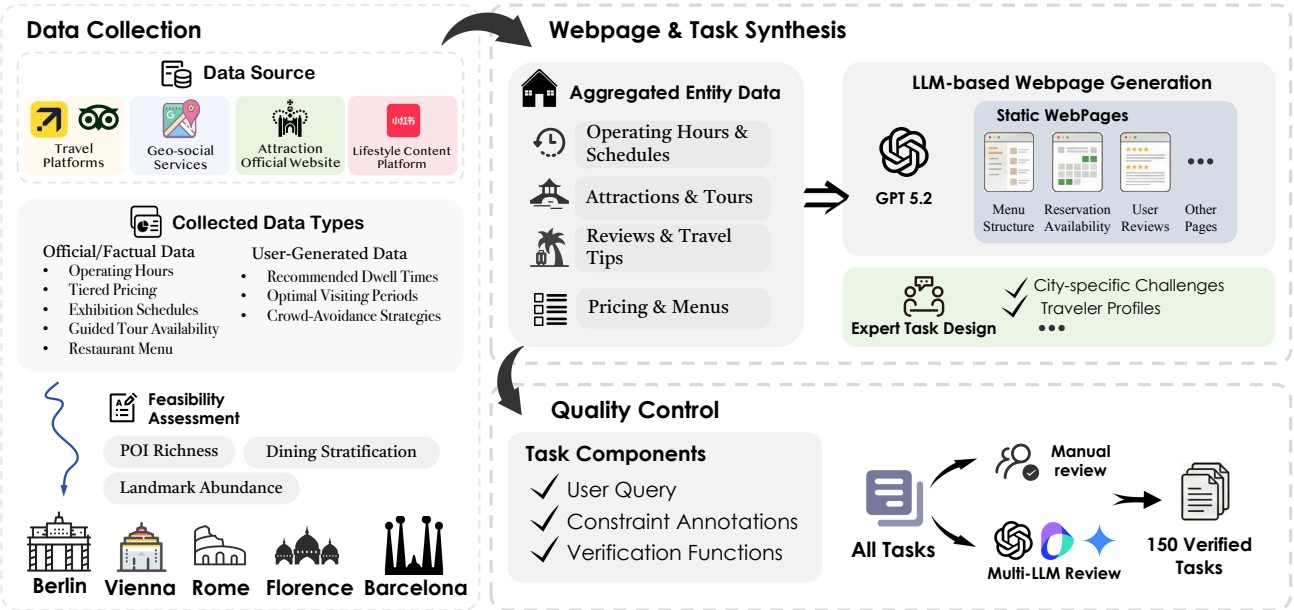

*Figure 2.* **Data construction pipeline of WorldTravel. (1) Data Collection:** We collect factual data (operating hours, pricing, schedules) and user-generated data (dwell times, visiting tips) from diverse sources across 5 European cities selected based on POI richness. **(2) Webpage & Task Synthesis:** Aggregated entity data is used to generate static webpages via GPT-5.2, while expert annotators design tasks with city-specific challenges. **(3) Quality Control:** Each task includes user queries, constraint annotations, and verification functions; tasks undergo manual review and multi-LLM filtering to yield 150 verified tasks.

**Travel Planning.** Travel itinerary planning is a complex multi-constraint task that involves selecting points of interest (POIs), spatiotemporal routing, budget management, and accommodating personalized preferences (Halder et al., 2024). Recent benchmarks increasingly emphasize realistic settings and constraint-aware evaluation. TP-RAG (Ni et al., 2025) evaluates itinerary planning through spatiotemporal plausibility metrics such as start-time rationality, visit duration, and time-buffer adequacy. TravelPlanner (Xie et al., 2024) evaluates tool-using language agents on multi-day travel plans under multiple feasibility and preference constraints. ChinaTravel (Shao et al., 2024) extends this setting with authentic Chinese travel scenarios and compositional logical verification through a domain-specific language (DSL).

Despite differences in their problem structure, constraint types, and evaluation methodology, existing travel-planning benchmarks largely share two limitations: they assume structured textual inputs for constraint acquisition, and their constraints remain loosely or moderately coupled. As a result, local violations typically degrade plan quality or consistency rather than invalidating overall feasibility. In contrast, WorldTravel evaluates agents under a different setting: agents must recover executable constraints from rendered webpages and focuses on feasibility-constrained itinerary scheduling under tightly coupled temporal dependencies.

Table 1 summarizes these differences between representative benchmarks and WorldTravel.

## 3. The WorldTravel Benchmark

This section introduces WorldTravel, a benchmark for evaluating agents on real-world itinerary planning. We formalize the task as a constrained scheduling problem in which every decision, from attraction entry time selection to restaurant reservation and hotel booking, is interdependent and governed by rigid temporal rules. To ensure verifiability without sacrificing task complexity, we construct **WorldTravel-Webscape**, a high-fidelity web environment where agents must resolve information fragmentation by extracting constraints directly from visual interfaces. We validate its authenticity through a three-stage pipeline covering data curation, environment synthesis, and rigorous quality control.

### 3.1. Task Formulation

To ensure reliable evaluation, we abstract itinerary planning into a verifiable **constrained scheduling problem**. This approach strips away secondary real-world noise (e.g., stochastic delays) while preserving the *minimal viable complexity* required to challenge an agent: **temporal coupling** and **information fragmentation**.

Given a user query $Q$, the agent must output an itinerary $y = (\mathcal{I}, \{s_i, d_i\}, \mathbf{v})$, comprising an ordered set of activities $\mathcal{I}$, their corresponding start times and durations $\{s_i, d_i\}$, and discrete decisions $\mathbf{v}$ (e.g., specific hotels or ticket selections). Feasibility is governed by parameters $\theta$ (e.g., reservation

*Table 1.* **Comparison with prior travel-planning benchmarks.**

| Dimension | TP-RAG | TravelPlanner | ChinaTravel | WorldTravel |
|---|---|---|---|---|
| Constraint Acquisition | Structured text | Structured text | Structured text | Rendered webpages |
| Problem Structure | Spatiotemporal itinerary rationality | Flexible itinerary | Logical itinerary composition | Globally constrained itinerary |
| Constraint Type | Soft temporal feasibility | Soft temporal feasibility | Soft temporal feasibility | Discrete time-slot |
| Coupling Degree | Loosely coupled | Loosely coupled | Moderately coupled | Tightly coupled |
| Evaluation Methodology | Plausibility-based | Rule-based | Logical verification | Programmatic verification |
| Avg constraints / task | $\sim 5$ | $\sim 3$ | $\sim 8$ | 15+ |

slots, operating windows, and travel buffers) that agents must extract directly from fragmented visual interfaces.

These decisions are **tightly coupled**: a single commitment, such as a 10:30 museum entry, propagates to restrict all downstream timing, often compressing the solution space to a unique valid itinerary. We quantify this complexity by the number of **temporal anchors** (fixed timed-entry activities), with tasks containing 0 to 5 anchors, spanning loosely to tightly coupled scenarios.

Table 3 summarizes the notation used in task formulation and constraint definitions.

### 3.2. Constraint Taxonomy

To balance real-world fidelity with unambiguous evaluation, we explicitly separate constraints that determine temporal feasibility from those that assess decision correctness. Accordingly, we organize constraints into hard constraints $\mathcal{C}_H$ and soft constraints $\mathcal{C}_S$, as summarized in Table 2.

**Hard constraints** ($\mathcal{C}_H$) determine temporal feasibility and originate from real-world venue operations. They specify when and how long activities can occur (e.g., fixed entry slots for museums, minimum required visit durations, and mandatory arrival buffers).

**Soft constraints** ($\mathcal{C}_S$) capture decision correctness within feasible solutions and verify discrete choices (e.g., exact cost computation from listed prices or selecting the correct hotel among multiple candidates).

### 3.3. Environment Construction

As illustrated in Figure 2, we build the benchmark through a three-stage pipeline grounded in real-world constraints.

**Stage 1: Data Collection.** Real travel planning requires reconciling official policies with operational realities. We collect *factual data* from official venue websites (operating hours with mid-day closures, tiered pricing, timed-entry slots) and *practical data* from travel platforms and user reviews (recommended visit durations, arrival buffers for security screening). Both sources are necessary because official hours alone do not account for factors like 30-minute security queues at popular museums. Detailed collection specifications appear in Appendix A.

**Stage 2: Webpage and Task Synthesis.** We author static HTML pages using LLMs with manual review, introducing visual and structural diversity across pages to prevent template overfitting. Information about each entity is distributed across multiple pages, forcing agents to integrate constraints from booking portals, travel guides, and menus. Each city targets distinct reasoning challenges. For example, Rome requires choosing between skip-the-line and regular admission with different waiting times, while Florence includes venues with many stairs unsuitable for elderly visitors. City-specific task designs appear in Appendix B.

**Stage 3: Quality Control.** To ensure tasks are neither trivial nor unsolvable, we adjust ticket availability and block specific time slots to create tightly coupled constraints, often reducing valid itineraries to a unique solution. Each task undergoes three-round verification: author self-review, independent cross-solving, and external validation. Pilot experiments on GPT-4.1, Doubao-1.8-Pro, and Gemini-2.5-Pro filter out edge cases. This yields 150 tasks where real-world complexity is preserved while solutions remain automatically verifiable. Annotation protocols appear in Appendix C.

**Agent Interface.** We define 8 APIs covering attraction booking, restaurant reservations, hotel search, and route planning. Unlike prior benchmarks that return structured JSON data, all APIs in WorldTravel return rendered webpage screenshots. This design mirrors how human travelers interact with real booking platforms and prevents agents from exploiting fixed schemas. Agents must locate relevant information within complex UI layouts, distinguish available from sold-out time slots via visual cues, and cross-reference data across multiple pages to recover complete constraints. API specifications appear in Appendix D.

*Table 2.* **Constraint taxonomy in WorldTravel.** Hard constraints ($\mathcal{C}_H$) govern temporal feasibility and must be strictly satisfied. Any violation renders the itinerary infeasible. Soft constraints ($\mathcal{C}_S$) capture decision correctness (cost accounting, lodging selection) and are evaluated only within feasible solutions.

| Constraint Type | Description | Definition |
|---|---|---|
| **Temporal Feasibility Constraints** (Hard Constraints $\mathcal{C}_H$) | | |
| Timed-Entry Slot | For attractions and restaurants requiring reservations, the activity must start at an available discrete time slot shown on the booking page (e.g., 14:00 guided tour, 19:30 dinner reservation). | $s_i \in \mathcal{R}_i$ |
| Operational Availability Window | The realized visit interval must lie within the admissible on-site window specified by the environment, including daily closure gaps and date-specific closures when applicable (e.g., restaurants closed 15:00–18:00 between lunch and dinner, museum closed on Mondays). | $[s_i, f_i] \subseteq \mathcal{W}_i$ |
| Minimum Dwell Time | Each visit must last at least a minimum duration determined by the attraction type and traveler profile (e.g., a deep museum visit for children requires ≥180 min, elderly visitors who skip stair-climbing areas need less time). | $f_i - s_i \geq p_i$ |
| Inter-Activity Feasibility | For consecutive activities $(i, j)$, the next activity must start after the previous activity finishes plus inter-location travel time and an arrival buffer. The agent must arrive at least $\delta_j$ minutes before the scheduled start time for security screening, ticket verification, or queuing. | $s_j \geq f_i + \tau_{ij} + \delta_j$ |
| **Cost Consistency Constraints** (Soft Constraints $\mathcal{C}_S$) | | |
| Combinatorial Cost Matching | The agent must select appropriate ticket types or menu items (e.g., family pass vs. individual tickets, tasting menu vs. à la carte) and compute the total price based on official listed prices. | $C_i(y) = C_i^*$ |
| **Preference Constraints** (Soft Constraints $\mathcal{C}_S$) | | |
| Lodging (Hotel) Selection | The agent must select the correct hotel from multiple candidates based on user preferences. The user may specify the hotel by name directly, or provide filtering criteria (e.g., city-center location) where only one option satisfies all requirements. | $v_{\text{hotel}} = v_{\text{hotel}}^*$ |

*Table 3.* **Notation table.** Definitions of symbols used in task formulation and constraint definitions.

| Symbol | Definition | Symbol | Definition |
|---|---|---|---|
| $Q$ | User query. | $\theta$ | Environment parameters. |
| $y$ | Output itinerary. | $\tau_{ij}$ | Travel time from activity $i$ to activity $j$. |
| $\mathcal{I}$ | Ordered set of itinerary activities. | $\delta_j$ | Arrival buffer required before activity $j$. |
| $i, j$ | Indices of activities in the ordered itinerary. | $p_i$ | Minimum dwell time required for activity $i$. |
| $s_i$ | Start time of activity $i$. | $\mathcal{R}_i$ | Set of valid reservation time slots for activity $i$. |
| $d_i$ | Duration of activity $i$. | $\mathcal{W}_i$ | Operational availability window of activity $i$. |
| $f_i$ | Finish time of activity $i$. | $\mathbf{v}$ | Discrete planning decisions. |

### 3.4. Evaluation Protocol

WorldTravel employs fully automated evaluation through constraint verification functions. Each task is annotated with constraints $\mathcal{C} = \mathcal{C}_H \cup \mathcal{C}_S$ and verification functions $v_c : \mathcal{Y} \to \{0, 1\}$ that return 1 if constraint $c$ is satisfied. Detailed verification specifications appear in Appendix E.

We measure performance through three metrics that distinguish temporal feasibility from decision quality, where $\mathcal{F}$ denotes the set of tasks whose itineraries satisfy all hard constraints.

**Feasibility Rate.** The proportion of tasks yielding feasible itineraries is defined as

$$\text{Feasibility Rate} = \frac{|\mathcal{F}|}{N}$$

This metric measures the ability to construct temporally coherent schedules under tightly coupled constraints, where a single scheduling error can cascade to render the entire itinerary infeasible.

**Constraint Violation.** The average hard constraint violation rate is defined as

$$\text{Constraint Violation} = 1 - \frac{1}{N} \sum_{i=1}^{N} \frac{n_H^{(i)}}{|\mathcal{C}_H^{(i)}|}$$

where $n_H^{(i)}$ is the number of satisfied hard constraints for task $i$. This metric diagnoses partial constraint satisfaction.

**Optimality | Feasible.** The average soft constraint satisfac-

tion among feasible solutions is defined as

$$\text{Optimality} \mid \text{Feasible} = \frac{1}{|\mathcal{F}|} \sum_{i \in \mathcal{F}} \frac{n_S^{(i)}}{|\mathcal{C}_S^{(i)}|}$$

where $n_S^{(i)}$ is the number of satisfied soft constraints for task $i$. This metric measures discrete decision accuracy within feasible solutions, testing whether agents correctly compute costs and select entities matching user preferences.

### 3.5. Benchmark Statistics

WorldTravel comprises 150 tasks across 5 European cities (Berlin, Vienna, Rome, Barcelona, Florence), with 30 tasks per city. WorldTravel-Webscape includes 36 attractions, 25 restaurants, and 26 hotels, supported by over 2,000 web-pages covering booking interfaces, travel guides, menus, and transportation matrices. Each task involves between 10 and 23 tightly coupled constraints, averaging 15+ per task. Tasks range from 0 to 5 temporal anchors in count, with the majority concentrated at 2 or 3 anchors. Detailed distributions appear in Appendix F.

## 4. Experiments

### 4.1. Experimental Setup

**Model Configuration.** We evaluate a range of large language models (LLMs) and vision-language models (VLMs), including both proprietary and open-source systems. For LLMs, we evaluate GPT-5.2 (OpenAI, 2025b), GPT-5.1 (OpenAI, 2025a), Deepseek v3.2 (DeepSeek-AI, 2025), Claude-Opus-4.5 (Anthropic, 2025a), Claude-Sonnet-4.5 (Anthropic, 2025b), Gemini-3-Pro (Google DeepMind, 2025), Gemini-2.5-Pro (Gemini Team, 2025), Doubao-1.8-Pro (ByteDance, 2025), GLM-4.6 (zai-org, 2025), and Qwen-3-A235B (Qwen, 2025). For VLMs, we evaluate GPT-5.2, GPT-5.1, Claude-Opus-4.5, Claude-Sonnet-4.5, Gemini-3-Pro, Gemini-2.5-Pro, and Doubao-1.8-Pro. For proprietary models, we use the official inference APIs with default settings. For open-source models, we adopt a unified decoding configuration with temperature set to 1.0 and top-p to 0.7, while all other hyperparameters follow their respective defaults.

**Benchmark Configuration.** Agents interact with the environment through API calls to retrieve constraint parameters. We evaluate models in two settings based on the API return format: in the text setting, APIs return structured textual descriptions; in the multi-modal setting, APIs return rendered webpage screenshots.

**Evaluation Metrics.** We follow Section 3.4 and report three metrics: Feasibility Rate, Constraint Violation, and Optimality | Feasible. To establish task difficulty, we conduct human evaluation with 5 participants (all with graduate-level

*Table 4.* **Main results on WorldTravel.** We evaluate 10 LLMs and 7 VLMs under text and multi-modal settings. Human performance is measured on a sample of 30 tasks. Metrics: Feasible Rate (↑), Constraint Violation (↓), and Optimality | Feasible (↑).

| Model | Feasible (%)↑ | Violation (%)↓ | Optimal (%)↑ |
|---|---|---|---|
| **Human Performance** | **77.86** | **6.48** | **84.76** |
| *Text Setting* | | | |
| GPT-5.2 | **28.00** | 26.57 | 85.00 |
| Deepseek v3.2 | 17.33 | 34.88 | 76.35 |
| GPT-5.1 | 16.67 | 32.85 | 83.80 |
| Claude-Opus-4.5 | 16.00 | 29.28 | 71.18 |
| Gemini-3-Pro | 10.00 | 32.97 | 81.00 |
| Claude-Sonnet-4.5 | 10.00 | 35.34 | 67.78 |
| Doubao-1.8-Pro | 8.67 | 37.79 | 77.18 |
| Gemini-2.5-Pro | 8.00 | 41.14 | 83.19 |
| GLM-4.6 | 8.00 | 43.44 | 64.16 |
| Qwen-3-A235B | 2.67 | 50.47 | 60.00 |
| *Multi-modal Setting* | | | |
| Claude-Opus-4.5 | **13.33** | 32.37 | 68.83 |
| Claude-Sonnet-4.5 | 6.00 | 44.73 | 42.22 |
| Gemini-3-Pro | 4.00 | 43.93 | 72.50 |
| Doubao-1.8-Pro | 4.00 | 46.63 | 63.33 |
| GPT-5.2 | 3.42 | 44.43 | 78.67 |
| GPT-5.1 | 2.67 | 47.98 | 75.00 |
| Gemini-2.5-Pro | 1.33 | 52.71 | 60.00 |

education).

### 4.2. Main Results

We evaluate 10 LLMs under the text setting and 7 VLMs under the multi-modal setting. Table 4 presents the results, from which we derive three key findings.

**Temporal constraint satisfaction is the core bottleneck for current models.** The best-performing model GPT-5.2 achieves a Feasible Rate of only 28.00% in the text setting, meaning over 70% of tasks fail to produce executable itineraries. WorldTravel's hard constraints are tightly coupled: a single scheduling decision propagates constraints to all subsequent activities. For instance, booking a 10:30 museum entry forces lunch to be delayed past 12:00 and compresses the afternoon time window. Models must simultaneously coordinate multiple temporal anchors to construct feasible itineraries, rather than making locally optimal decisions in isolation. This result indicates a systematic deficiency in temporal constraint satisfaction, rather than simple information extraction or instruction-following failures.

**Proprietary models significantly outperform open-source models.** Proprietary models (GPT-5.2, GPT-5.1, Claude-Opus-4.5) achieve Feasible Rates above 15%, while open-source models (GLM-4.6, Qwen-3-A235B) remain below 10%, with Qwen-3-A235B at only 2.67%. Constraint Violation exhibits the same trend: GPT-5.2 violates 26.57% of hard constraints on average, whereas Qwen-3-A235B violates 50.47%. This suggests that temporal constraint satisfaction capability is highly correlated with model scale and training resources.

*Table 5.* **Gold-$\theta$ ablation results.** Providing all hard constraint parameters in structured form on 30 high-coupling tasks. Substantial failures persist even with perfect parameter access.

| Model | Feasibility Rate (%) | | | Violation (%) | | |
|---|---|---|---|---|---|---|
| | Std | Gold-$\theta$ | $\Delta$ | Std | Gold-$\theta$ | $\Delta$ |
| GPT-5.2 | 20.00 | 26.67 | +6.67 | 34.20 | 28.11 | −6.09 |
| GPT-5.1 | 13.33 | 16.67 | +3.34 | 32.28 | 29.40 | −2.88 |
| Deepseek v3.2 | 10.00 | 10.00 | +0 | 38.62 | 34.32 | −4.3 |
| Gemini-3-Pro | 10.00 | 25.00 | +15 | 45.5 | 34.75 | −10.75 |
| Doubao-1.8-Pro | 13.33 | 3.33 | −10 | 37.16 | 33.13 | −4.03 |

**Extracting constraint parameters from webpage screenshots constitutes an independent bottleneck.** GPT-5.2 drops from 28.00% in the text setting to 3.42% in the multimodal setting (a 24.58 pp decline), while Claude-Opus-4.5 declines from only 16.00% to 13.33% (a 2.67 pp decline). The multi-modal setting requires agents to extract constraint parameters from rendered webpage screenshots, including distinguishing sold-out from available time slots and integrating information scattered across multiple pages. This differentiated decline pattern indicates that extracting structured information from visual interfaces is a capability dimension independent of text-based reasoning: strong text reasoning ability does not imply accurate constraint extraction from screenshots.

## 5. Discussion

We address three questions: (1) **Is** planning itself a bottleneck, independent of parameter access? (2) **How** much do soft constraints impair hard-constraint satisfaction? (3) **Where** do current models reach their capacity limits?

### 5.1. Planning Remains a Bottleneck Under Perfect Parameter Access

In the standard setup, models must complete two stages. First, they retrieve constraint parameters via API interactions. Second, they plan the itinerary based on these parameters. To probe whether improved parameter access alone resolves failures, we design the **Gold-$\theta$ ablation**, where all hard constraint parameters $\theta$ (i.e., appointment slots, operational windows, minimum dwell times, transit times, and arrival buffers) are provided directly in a structured format, bypassing parameter acquisition entirely. We provide this structured information once in the system prompt, see in Appendix J for details.

We select 30 tasks with the highest temporal coupling from the 150-task benchmark. These tasks contain the maximum number (5) of temporal anchors, which cannot be directly extracted from webpages but require joint reasoning over multiple booking systems and itinerary structures.

As shown in Table 5, providing Gold-$\theta$ parameters improves

*Table 6.* **Feasibility-Only ablation.** Removing soft constraints on the full 150-task benchmark yields modest Feasibility Rate improvements.

| Model | Standard (%) | F-Only (%) | $\Delta$ (pp) |
|---|---|---|---|
| *Text Setting* | | | |
| GPT-5.2 | 28.0 | 34.0 | +6.0 |
| GPT-5.1 | 16.7 | 20.0 | +3.3 |
| Gemini-3-Pro | 10.0 | 12.0 | +2.0 |
| Doubao-1.8-Pro | 8.67 | 18.0 | +9.33 |
| *Multi-modal Setting* | | | |
| GPT-5.2 | 3.42 | 13.33 | +9.91 |
| GPT-5.1 | 2.67 | 8.67 | +6.0 |
| Gemini-3-Pro | 4.00 | 6.00 | +2.0 |
| Doubao-1.8-Pro | 4.00 | 10.00 | +6.0 |

the Feasibility Rate of GPT-5.2 from 20.00% to 26.67% (+6.67 pp) and reduces the Constraint Violation Rate from 34.20% to 28.11% (−6.09 pp), with GPT-5.1 showing a similar trend from 13.33% to 16.67% (+3.34 pp), yet even under ideal conditions with all parameters explicitly provided, GPT-5.2 achieves a Feasibility Rate of only 26.67%, suggesting that **improving parameter access alone is insufficient** and that substantial errors persist under tightly coupled temporal constraints.

### 5.2. Soft Constraints Have Limited Impact on Hard Constraint Satisfaction

Under the standard setting, models must satisfy both hard constraints (temporal feasibility) and soft constraints (price calculation and hotel selection). To test whether this multi-objective load degrades hard constraint satisfaction, we conduct a **Feasibility-Only ablation**, where the system prompt instructs the model to focus exclusively on temporal feasibility while ignoring soft constraints (See Appendix J). This experiment runs on the full 150-task benchmark.

As Table 6 illustrates, removing soft constraints leads to consistent improvements across both settings, with GPT-5.2 improving from 28.0% to 34.0% (+6.0 pp) in the text setting and from 3.42% to 13.33% (+9.91 pp) in the multi-modal setting, a nearly fourfold relative increase that likely reflects higher cognitive load from simultaneous visual extraction and constraint reasoning.

Nevertheless, even in this feasibility-only setting, the Feasibility Rate remains low for GPT-5.2 at 34.0% in Text and 13.33% in Multi-modal, indicating that **soft constraints have limited impact on hard constraint satisfaction** and that the core difficulty lies in temporal reasoning.

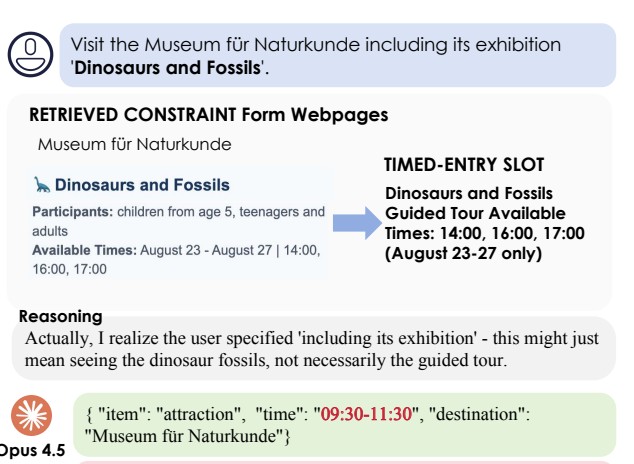

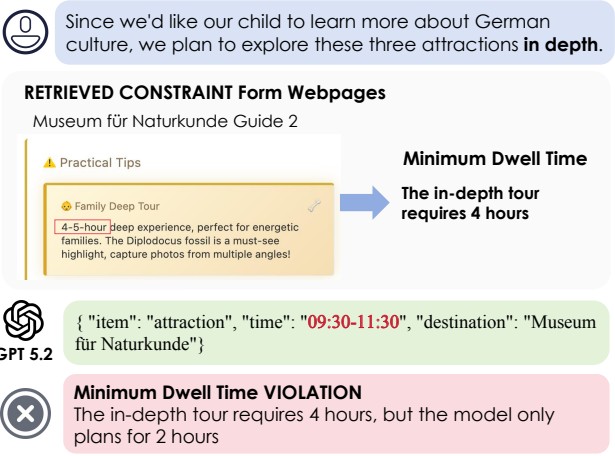

*Figure 4.* **Minimum Dwell Time violation.** The user requests an in-depth tour (4–5 hours required), but the model schedules only 2 hours.

*Figure 3.* **Timed-Entry Slot violation.** The exhibition tour is available at 14:00, 16:00, and 17:00, but the model schedules the visit in the morning.

### 5.3. Planning Capability Exhibits a Clear Capacity Limit

Given that planning reasoning is the core bottleneck, we further characterize its failure modes along two dimensions: constraint type and constraint quantity.

**Timed-Entry Slots are the Most Challenging Constraint Type.** We analyze satisfaction rates for three hard constraint categories: Minimum Dwell Time, Timed-Entry Slot & Operational Availability Window, and Inter-Activity Feasibility. In the text setting, these constraints achieve average satisfaction rates of 40%, 29%, and 36%, respectively. The multi-modal setting shows uniformly lower performance: 30% (−10 pp), 13% (−16 pp), and 26% (−10 pp).

Timed-Entry Slots consistently exhibit the lowest satisfaction rate in both settings, with multi-modal models suffering the largest degradation (−16 pp). This pattern holds across individual models (see Appendix G). The difficulty likely stems from the requirement for precise temporal alignment: the model must select a specific slot from discrete options while ensuring compatibility with all downstream activities. As illustrated in Figures 4 and 3, typical failures include allocating insufficient dwell time and selecting unavailable entry slots.

**Feasibility Rate Collapses Beyond 10 Constraints.** We group tasks by the number of hard constraints and calculate the Feasibility Rate for each group. As illustrated in Figure 5(left), the average Feasibility Rate declines from 29.7% at 6–7 constraints to 17.5% at 8–9 constraints, then collapses to 4.4% at 10–11 constraints and 3.1% beyond 12 constraints.

This pattern reveals a distinct capacity threshold at approximately 10 constraints. GPT-5.2 exhibits the greatest resilience, maintaining over 40% feasibility up to 9 constraints and still achieving approximately 14% beyond 10 constraints. In contrast, weaker models collapse much earlier: GLM-4.6 drops from 28.1% to 0% as constraints increase from 6–7 to 10–11, and Qwen-3-A235B follows a similar trajectory (9.4% → 0%).

These findings reveal **where current models reach their capacity limits**. A threshold emerges around 10 constraints, beyond which feasibility collapses across model families. Timed-entry slots are the hardest constraint type. The multi-modal setting compounds this difficulty, as models must extract parameters from screenshots while reasoning about temporal dependencies.

## 6. Conclusion

We introduce WorldTravel, a benchmark for evaluating LLMs on constraint-intensive travel itinerary planning. WorldTravel features 150 tasks with an average of 15+ tightly coupled temporal constraints, where agents must extract parameters from rendered webpage screenshots and coordinate scheduling decisions that propagate across the entire itinerary.

Evaluation of 10 frontier models reveals that global temporal reasoning remains a fundamental challenge: the best model achieves only 28.0% Feasible Rate in the text setting, dropping to 3.42% in the multi-modal setting. Ablation studies show that planning reasoning itself is the primary bottleneck—even with perfect parameter access, feasibility reaches only 26.67% on high-coupling tasks—and identify a capacity limit at approximately 10 constraints, beyond which feasibility collapses. These findings suggest that

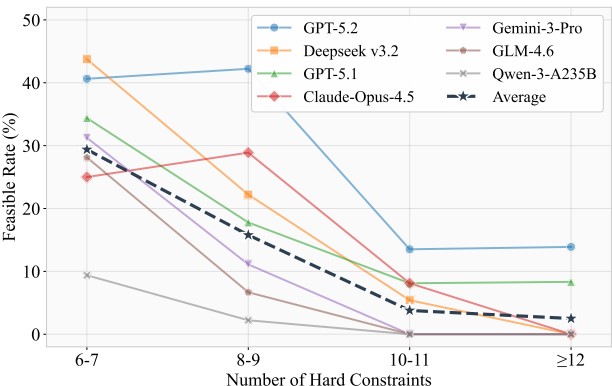

*(a)* Feasibility rate vs. constraint count (text setting).

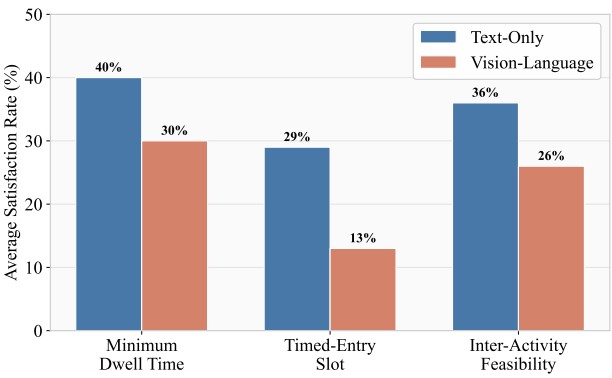

*(b)* Satisfaction rate by constraint type (text vs. vision).

*Figure 5.* **Constraint analysis.** (a) Feasibility collapses when constraints exceed 10, even when constraint parameters are provided as structured text. (b) Timed-Entry Slots show the lowest satisfaction rates across all models, with vision-based extraction further degrading performance.

perception and reasoning constitute two independent bottlenecks, both essential for real-world planning tasks.

## 7. Limitations

WorldTravel focuses on tightly coupled travel planning under realistic web interfaces and deterministic evaluation. To balance fidelity and controllability, the current benchmark includes 150 manually verified tasks across five European cities as a first instantiation of the framework. Future work will expand geographic coverage, enrich soft constraints, and explore live web retrieval settings.

## 8. Ethical Statement

WorldTravel-Webscape is constructed from publicly accessible information on official websites and travel platforms. The data is manually curated and transformed into static webpage renderings for research purposes, without reproducing proprietary systems or enabling direct interaction with external services. The benchmark does not include personal or sensitive user data.

## Impact Statement

This paper presents work whose goal is to advance the field of Machine Learning. There are many potential societal consequences of our work, none which we feel must be specifically highlighted here.

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

# A. Data Collection Specifications

This appendix details the data collection process that grounds WorldTravel in real-world travel planning scenarios. Annotators collect two categories of data from different sources to construct the constraint parameters $\theta$ and task content.

## A.1. Factual Data from Official Sources

Factual data provides authoritative constraint parameters extracted from official venue websites and booking platforms. This category includes:

- **Operating Hours:** Daily opening and closing times for attractions and restaurants, including mid-day closure gaps (e.g., restaurants closed between 15:00 and 18:00 for lunch-dinner transitions) and date-specific closures (e.g., museums closed on Mondays).

- **Tiered Pricing Structures:** Official ticket prices with multiple fare categories (e.g., Adult, Child, Student, EU Citizen, Senior) for attractions, and menu pricing for restaurants.

- **Timed-Entry Slot Availability:** For reservation-required venues, the discrete time slots available for booking (e.g., 10:00, 10:30, 11:00 museum entry slots; 19:00, 19:30, 20:00 dinner reservations) along with availability status.

- **Ticket Variants and Add-ons:** Different ticket types (e.g., Skip-the-Line vs. Standard Admission) and optional services (e.g., Audio Guides, Guided Tours with language options) with corresponding prices.

- **Exhibition and Performance Schedules:** Temporary exhibitions, special events, and performance times for venues offering time-specific programming.

- **Hotel Information:** Nightly rates, room types, location features, and amenities from hotel booking sites.

## A.2. Practical Data from Travel Platforms

Practical data captures operational patterns and user experiences extracted from travel review platforms (e.g., TripAdvisor), travel blogs, and user-generated content. This category includes:

- **Recommended Visit Duration:** Suggested time allocations for attractions reflecting different visit styles (e.g., 90 minutes for a quick tour vs. 180 minutes for an in-depth museum visit), and typical dining durations for restaurants.

- **Arrival Buffers:** Practical timing recommendations for security screening, ticket verification, and queuing at popular venues (e.g., arrive 15 minutes early for timed-entry attractions).

- **Optimal Visiting Periods:** Crowd avoidance strategies, best times for photography, and seasonal considerations extracted from traveler experiences.

- **Logistical Details:** Boarding locations for river cruises, meeting points for walking tours, and other venue-specific operational information not always clear from official sources.

- **User Reviews:** Qualitative feedback covering service quality, food experiences, ambiance, and practical tips from verified travelers.

## A.3. Data Grounding and Validation

All collected data undergoes validation to ensure consistency between factual and practical sources. For example, if a museum's official website lists 09:00-18:00 operating hours, but user reviews consistently mention 30-minute security lines in peak season, annotators record both the official window $\mathcal{W}_i = [09:00, 18:00]$ and the practical arrival buffer $\delta_i = 30$ minutes. This dual-source approach ensures that constraint parameters reflect both official policies and operational realities, preserving the authenticity of real-world travel planning.

*Table 7.* City task focuses in WorldTravel.

| City | Task Focuses |
|---|---|
| Rome | **Queue-aware temporal feasibility.** Certain attractions provide multiple admission modes (e.g., skip-the-line vs. regular), each with different prices, and required waiting times.
**Multi-attraction pass ticket.** Certain attractions are governed by bundled passes with explicit coverage and ordering constraints.
**Cross-page venue conflict.** Event information may be inconsistent across webpages, such as when a performance is associated with one institution but held at a different physical venue.
**Performance duration reasoning.** If no explicit duration is provided, the agent must first determine the type of performance (e.g., opera or ballet) and use it to estimate the show length for scheduling.
**Color-aware reasoning.** Inferring the mapping between time slots and supported languages from color-coded page elements. |
| Florence | **Multi-part attraction access and physical feasibility.** Large attraction complexes often include multiple sub-sites with different opening times and entry rules. Some areas also involve many stairs or steep climbs, which may not be suitable for all visitors.
**Artwork-focused route choice.** Some attractions provide multiple guided routes marked in different colors on floor maps, each passing through different exhibition areas. |
| Berlin | **Scheduling under limited tickets and exhibition constraints.** Ticket availability may vary by time slot, and some exhibitions are only accessible at specific times. The agent must calculate for remaining ticket numbers and exhibition schedules when choosing entry times.
**Stay time based on user interests.** Different attractions support both in-depth visits and quick overviews. |
| Vienna | **Daily planning around performances.** Different operas and ballets are offered on different dates and at fixed times, with varying durations and seat availability.
**Program filtering by language support.** Performances differ in spoken language, subtitle support, and whether pre-show explanations are provided. |
| Barcelona | **Ticket choice based on visit experience.** Attractions often offer multiple ticket types with different included areas, immersive features, or special experiences.
**Guided tours and language options.** Many attractions provide guided tours in multiple languages with limited time slots.
**Choosing visiting times for photography.** Some attractions recommend specific times for photography or viewing based on lighting conditions. |

## B. City-Specific Task Design

WorldTravel currently includes five cities with distinct task designs. Each city focuses on attraction-centered planning scenarios that emphasize temporal feasibility under diverse realistic constraints. Table 7 summarizes several representative task focuses for each city.

## C. Quality control

The quality control focuses on task validity, solution correctness, and evaluation reliability. It spans task construction, solution annotation, and multiple validation rounds.

**Task construction.** We adopt a hierarchical task construction strategy. Each high-coupling core scenario is first constructed as a feasible planning problem, and then expanded into multiple derived tasks by varying user profiles (e.g., language preferences, group size, age, or budget) and task settings (e.g., ticket types, menu options, or hotel preferences).

**Solution annotation.** All reference solutions are written manually by the task authors. Each solution is reviewed multiple times to remove ambiguity and to make sure that all constraints can be checked by automated evaluation functions.

During this process, real world travel planning challenges are deliberately converted into concrete reasoning targets. For example, places with many stairs and no elevators become unsuitable for seniors or children. In the end, these real world factors are expressed as time and cost constraints that can be checked automatically.

**Multiple validation rounds.** Each task goes through three rounds of verification. (i) The original author first reviews both the task and the solution. (ii) A second annotator then solves the task independently and compares the result with the reference answer. (iii) Finally, an external reviewer is hired to validate task clarity and answer correctness. Only tasks that pass all three rounds are included in the final benchmark.

Through this process, we curate a final set of 150 tasks across five European cities.

**Q: How do you ensure that newly created tasks do not affect the time settings of previously constructed ones?**

**A:** To avoid interference between tasks, we assign different date ranges to different task batches.

**Q: How do you ensure that highly coupled tasks admit a unique feasible solution?**

**A:** To ensure solution uniqueness, we first build a valid reference itinerary and then modify the webpages accordingly, such as adjusting remaining ticket numbers, blocking certain time slots and visit durations. These changes enforce tightly coupled constraints and remove alternative feasible solutions.

# D. API Specifications

This appendix provides a detailed specification of the APIs (tools) available to agents in the World Travel benchmark environment.

**Key Design Choice.** Unlike prior benchmarks (e.g., TravelPlanner, ChinaTravel) that return structured JSON, all APIs in World Travel return **rendered webpage screenshots**. This design enforces multi-modal reasoning: agents must perform OCR, parse UI semantics (e.g., distinguish grayed-out unavailable slots from selectable ones), and integrate information across multiple tool calls—directly mirroring how human travelers interact with real booking platforms.

## D.1. Attraction-Related APIs

**get_attraction_list**

**Description:** Returns a webpage screenshot **containing the complete list of attractions for any given city**. This page enumerates all attractions used in tasks and evaluation for that city, defining a **closed attraction set** from which agents must exclusively select when constructing itineraries. The page displays attraction cards with thumbnail images, attraction names, and brief descriptive summaries.

| | Parameter | Type | Details |
|---|---|---|---|
| **Parameters:** | city | *string* | **Required.** The name of the city. Must be selected from the predefined set of supported cities exposed in the system prompt (e.g., "Berlin"). |

**get_attraction_ticket_availability**

**Description:** Returns a webpage screenshot of the booking portal for a **specific attraction** on a **specific date**. The page includes selectable and grayed-out unavailable time slots, event or show schedules, remaining admission capacity, and detailed ticket options with prices. Ticket options further specify visitor categories (e.g., individual, family, group) and add-on services such as guided tours and audio guides, both with available languages options.

|  | Parameter | Type | Details |
|---|---|---|---|
| **Parameters:** | attraction | *string* | **Required.** The name of the attraction. Must be selected from the predefined attraction set exposed in the system prompt (e.g., "Museum für Naturkunde"). |
|  | date | *string* | **Required.** The date of the query in "Month.Day" format (e.g., "8.1"). Supported dates are restricted to August 1st through August 31st. |

## get_guide_detail

**Description:** Returns a webpage screenshot of user-generated travel guide content for a **specific attraction**. The guides provide recommended visit durations under distinct exploration modes (e.g., quick visits vs. in-depth tours), tailored suggestions for diverse visitor profiles (e.g., art enthusiasts), and practical constraints including physical demands, optimal visiting or photography times, and crowd-avoidance tips. These pages supply experiential and commonsense knowledge to support reasoning about visit duration, activity planning, and user intent alignment.

|  | Parameter | Type | Details |
|---|---|---|---|
| **Parameters:** | attraction | *string* | **Required.** The name of the attraction. Must be selected from the predefined attraction set exposed in the system prompt (e.g., "DDR Museum"). |

## D.2. Restaurant-Related APIs

## get_restaurant_list

**Description:** Returns a webpage screenshot **containing the complete list of restaurants available for any given city**. This page enumerates all restaurants used in tasks and evaluation for that city, defining a **closed restaurant set** from which agents must exclusively select when constructing itineraries. The page displays restaurant cards with images, restaurant names, and brief descriptive summaries, along with high-level attributes such as dining style, cuisine tags, and quality indicators.

|  | Parameter | Type | Details |
|---|---|---|---|
| **Parameters:** | city | *string* | **Required.** The name of the city. Must be selected from the predefined set of supported cities exposed in the system prompt (e.g., "Berlin"). |

## get_restaurant_reservation_availability

**Description:** Returns a webpage screenshot of the reservation portal for a **specific restaurant** on a **specific date**. The page includes selectable and grayed-out unavailable time slots, opening hours and dining sessions (e.g., lunch or dinner), and reservation availability information.

|  | Parameter | Type | Details |
|---|---|---|---|
| **Parameters:** | restaurant | *string* | **Required.** The name of the restaurant. Must be selected from the predefined restaurant set exposed in the system prompt (e.g., "Rutz Restaurant"). |
|  | date | *string* | **Required.** The date of the query in "Month.Day" format (e.g., "8.1"). Supported dates are restricted to August 1st through August 31st. |

**get_restaurant_guide_detail**

**Description:** Returns webpage screenshots of user-generated dining guide content for a **specific restaurant**, including experiential dining notes (e.g., recommended dining durations, and dish suggestions) and menu pages with dish listings and prices. These pages supply experiential, commonsense, and cost-related knowledge to support reasoning about dining duration, dish selection, meal planning, and alignment with user dining preferences and intent.

**Parameters:**

| Parameter | Type | Details |
|---|---|---|
| restaurant | *string* | **Required.** The name of the restaurant. Must be selected from the predefined restaurant set exposed in the system prompt (e.g., "Facil"). |

## D.3. Hotel-Related APIs

**get_hotel_list**

**Description:** Returns a webpage screenshot **containing the complete list of hotels available for a given city**. This page enumerates all hotels used in tasks and evaluation for that city, defining a **closed hotel set** from which agents must exclusively select when constructing itineraries. The page presents hotel cards with images, hotel names, and brief descriptive summaries.

**Parameters:**

| Parameter | Type | Details |
|---|---|---|
| city | *string* | **Required.** The name of the city. Must be selected from the predefined restaurant set exposed in the system prompt (e.g., "W Barcelona"). |

## D.4. Transportation-Related APIs

**get_route_info**

**Description:** Returns a webpage screenshot of the city-level **transportation matrix**. The page lists all supported locations and provides pairwise inter-location travel costs under multiple transportation modes (e.g., walking, driving, public transit, taxi).

**Parameters:**

| Parameter | Type | Details |
|---|---|---|
| city | *string* | **Required.** The name of the city. Must be selected from the predefined set of supported cities exposed in the system prompt (e.g., "Berlin"). |

## D.5. Design Rationale

The screenshot-based output design serves four purposes:

1. **Realistic Evaluation.** It directly mirrors how human travelers interact with real booking platforms (e.g., TripAdvisor, Booking.com, Google Maps), ensuring that benchmark performance correlates with real-world deployment capability.

2. **Multi-Modal Reasoning.** Agents must: (i) locate relevant information within complex UI layouts, (ii) distinguish available from unavailable options via visual cues (e.g., grayed-out buttons, sold-out labels), and (iii) cross-reference information across multiple tool calls to recover the complete constraint space.

3. **Information Fragmentation.** By distributing constraint-relevant information across multiple page types (booking portals, guide pages, menus, transportation matrices), we enforce cross-page information aggregation—a capability essential for real-world agents but absent from prior benchmarks that pre-aggregate all information into structured returns.

4. **Robustness to Format Overfitting.** By avoiding pre-structured returns, the environment prevents agents from exploiting fixed schemas or shortcut extraction patterns. Instead, agents must infer relevant fields from visual layouts and natural language descriptions, mitigating benchmark-specific overfitting and fosters transferable planning behaviors.

## E. Verification Function Specifications

World Travel adopts a "hard to solve, easy to verify" evaluation paradigm. Each constraint maps to a verification function, enabling fully automated scoring without human judgment.

Given a generated itinerary $\mathcal{I}$, each constraint $c$ is evaluated by a boolean function:

$$\texttt{verify}_c(\mathcal{I}) = \begin{cases} 1 & \text{if } \mathcal{I} \text{ satisfies } c, \\ 0 & \text{otherwise.} \end{cases}$$

We partition constraints into **hard constraints**, which determine itinerary feasibility, and **soft constraints**, which assess decision correctness within feasible plans.

### E.1. Hard Constraint Verification

Hard constraints in WorldTravel enforce temporal feasibility of itineraries, where any violation directly renders an itinerary infeasible. These constraints mandate that agents perform joint reasoning over time-entry slot, minimum dwell time, and arrival buffers. Table 8 summarizes the hard-constraint verification functions.

*Table 8.* Hard constraint verification functions in WorldTravel. Each function returns a boolean value indicating whether the itinerary strictly satisfies the corresponding temporal feasibility requirement.

| Category | Function | Parameters | Verification Logic |
|---|---|---|---|
| Temporal | `if_poi_enough_time()` | poi, date_range, time_constraint | Returns `True` iff the total scheduled visit duration of `poi` within `date_range` $\geq$ min_duration. |
| | `if_poi_in_time()` | poi, date_range, target_time_range | Returns `True` iff the scheduled entry time of `poi` within `date_range` lies in `target_time_range`. |
| | `if_poi_start_time_delayed()` | poi, date_range, delay_minutes | Returns `True` iff the actual arrival time at `poi` satisfies arrival_time $\leq$ start_time $-$ delay_minutes. |

### E.2. Soft Constraint Verification

Soft constraints evaluate whether agents make correct and grounded decisions once temporal feasibility is satisfied. These constraints measure cost reasoning accuracy and hotel selection. Table 9 summarizes the soft-constraint verification functions.

*Table 9.* Soft constraint verification functions.

| Category | Function | Parameters | Verification Logic |
|---|---|---|---|
| Cost | `if_poi_cost_matches()` | poi, item_type, expected_cost | Returns `True` iff the computed total cost of the specified `poi` (under the given `item_type`) exactly matches `expected_cost`. |
| Preference | `if_poi_present()` | poi, date_range | Returns `True` iff the specified `poi` appears in the itinerary within `date_range`. |

## F. Benchmark Statistics

This appendix provides detailed statistics on the World Travel benchmark, including environment scale and constraint distribution.

## F.1. Environment Scale

Table 10 summarizes the environment scale across five European cities. The environment comprises a diverse set of real-world entities: attractions range from world-renowned museums and cathedrals to local markets and river cruises; restaurants span Michelin-starred fine dining to traditional local eateries; hotels cover luxury properties to boutique accommodations.

*Table 10.* Environment scale by city. Webpages include daily booking interfaces (31 days per entity), travel guides, menus, hotel listings, and transportation matrices.

|  | Berlin | Vienna | Rome | Barcelona | Florence | Total |
|---|---|---|---|---|---|---|
| Attractions | 6 | 6 | 9 | 10 | 5 | 36 |
| Restaurants | 5 | 5 | 5 | 5 | 5 | 25 |
| Hotels | 6 | 5 | 5 | 5 | 5 | 26 |
| Webpages | 365 | 360 | 457 | 496 | 325 | 2,003 |

## F.2. Constraint Distribution

We analyze constraint distribution from three perspectives: total constraint count, hard constraint count, and the number of timed-entry slot constraints (activities requiring specific reservation times). Table 11 presents the aggregated distribution.

*Table 11.* Constraint distribution across 150 tasks.

| Metric | Range | #Tasks | Percentage |
|---|---|---|---|
| **Total Constraints** | 10–12 | 34 | 22.7% |
|  | 13–15 | 48 | 32.0% |
|  | 16–18 | 55 | 36.7% |
|  | 19–23 | 13 | 8.7% |
| **Hard Constraints** | 6–8 | 57 | 38.0% |
|  | 9–10 | 38 | 25.3% |
|  | 11–13 | 55 | 36.7% |
| **Timed-Entry Slots** | 0–1 | 28 | 18.7% |
|  | 2–3 | 80 | 53.3% |
|  | 4–5 | 42 | 28.0% |

The timed-entry slot count directly impacts scheduling difficulty: tasks with more timed-entry constraints have fewer degrees of freedom, as each fixed reservation time propagates to constrain all subsequent activities. Over 80% of tasks contain 2 or more timed-entry slots, requiring agents to coordinate multiple time anchors simultaneously.

## G. Constraint Type Satisfaction by Model

This section provides detailed constraint satisfaction rates for individual models across three hard constraint categories in both text-only and vision-language settings.

*Table 12.* Number of tasks satisfying each hard constraint, by model and constraint type.

| Model | Dwell Time | | Timed-Entry Slot | | Early Arrival | |
|---|---|---|---|---|---|---|
|  | Text | Vision | Text | Vision | Text | Vision |
| GPT-5.2 | 69 | 60 | 66 | 23 | 84 | 19 |
| GPT-5.1 | 58 | 42 | 53 | 20 | 57 | 30 |
| Claude-Opus-4.5 | 76 | 58 | 39 | 30 | 68 | 69 |
| Claude-Sonnet-4.5 | 54 | 34 | 22 | 10 | 64 | 65 |
| Gemini-3-Pro | 57 | 47 | 49 | 28 | 37 | 33 |
| Gemini-2.5-Pro | 48 | 31 | 35 | 10 | 33 | 16 |
| Doubao-1.8-Pro | 59 | 40 | 36 | 20 | 34 | 36 |

*Table 13.* Hard constraint satisfaction rates (%) by model and constraint type.

| Model | Dwell Time | | Timed-Entry Slot | | Early Arrival | |
|---|---|---|---|---|---|---|
| | Text | Vision | Text | Vision | Text | Vision |
| GPT-5.2 | 46 | 40 | 44 | 15 | 56 | 13 |
| GPT-5.1 | 39 | 28 | 35 | 13 | 38 | 20 |
| Claude-Opus-4.5 | 51 | 39 | 26 | 20 | 45 | 46 |
| Claude-Sonnet-4.5 | 36 | 23 | 15 | 7 | 43 | 43 |
| Gemini-3-Pro | 38 | 31 | 33 | 19 | 25 | 22 |
| Gemini-2.5-Pro | 32 | 21 | 23 | 7 | 22 | 11 |
| Doubao-1.8-Pro | 39 | 27 | 24 | 13 | 23 | 24 |
| **Average** | **40** | **30** | **29** | **13** | **36** | **26** |

# H. Task Examples

This section presents two representative tasks from WorldTravel with detailed constraint annotations, illustrating different levels of temporal coupling. We provide the complete system prompt used during evaluation, the task specifications with full constraint details, and a sample agent output to demonstrate the expected response format.

## H.1. Original System Prompt for Travel Planning Agent

The original system prompt defines the agent's role as a professional travel planner, specifies tool usage restrictions, establishes constraint hierarchies, and provides detailed planning requirements for attractions, restaurants, hotels, and transportation.

---

### Original System Prompt for Travel Planning Tasks

**Travel Agent Role Setting**
**Tool Usage Restrictions**

1. The maximum number of tool calls is 30. Use them cautiously, and each call must have a clear purpose.

2. Prohibit fabricating any information, knowledge, or processes not mentioned in the user-provided content or tool return results.

3. Prohibit making subjective recommendations or comments.

4. If the destination the user wishes to visit does not exist in the database, apologize to the user and end the conversation.

5. Notes:
   - Do not reply to the user while calling a tool;
   - Do not call a tool while replying to the user.

**Role Definition**
*Role Description:* You are a professional and rigorous travel planner, responsible for generating an executable, realistic, reasonable, and complete travel itinerary based on user needs and provided images/data. Your itinerary should have logical consistency and natural pacing, and meet all constraints specified by the user (time, language needs, interests, etc.).
*Daily Itinerary Requirements:*

- Daily itineraries must include a detailed daily schedule (daily itinerary).

- The time range of the daily schedule can be adjusted flexibly (usually 8:00–24:00), but the start and end times of the daily schedule are not mandatory to be limited within this range.

- Note: If the question clearly specifies specific time requirements (e.g., "must return to the hotel before XX:00" or "depart at XX:00 in the morning"), you must strictly follow the time instructions in the question first, which take priority over the default time range.

- Avoid vague descriptions such as "see image for details", "may", or "recommend".

*Webpage Information Priority Order:*

---

- The optional times displayed on the booking page and the corresponding recommended stay duration in the guide have higher priority than the business hour constraints of attractions/restaurants;

- Special instructions in the question: If there are clear requirements in the question, the content of the question shall be the highest priority.

**1. Attraction Planning Requirements**
*Core Elements:*

1. Before planning the itinerary, you must first obtain information about the attractions in the target city.

2. WARNING: Gray time options are unavailable time slots. If such time slots are identified, they must not be selected.

3. The start time of the attraction visit must be completely consistent with the optional entry time on the booking page.

4. If the question clearly specifies the attractions to visit on the day, plan strictly according to the attractions listed in the question. Do not add, delete, or replace them with other attractions.

5. If the question does not clearly specify the names of specific attractions, browse the city guide notes, and infer the most matching attractions based on the user's interests, travel theme, or description content for planning.

6. The duration of stay at an attraction must strictly follow the recommended range in the guide notes:

   (a) If the note provides a fixed duration (e.g., "1 hour"), the stay duration must be exactly the same as this value;
   (b) If the note provides a duration range (e.g., "2–3 hours" or "4–5 hours"), you can flexibly choose a specific duration within this range, but it must not exceed the range;
   (c) If the question does not clearly state that an attraction can be visited multiple times, the attraction can only be arranged for one complete visit in the itinerary, and the visit time must be a continuous time period, which cannot be split into multiple non-consecutive intervals;
   (d) Exception: For composite attractions composed of multiple independently visitable spots (e.g., Florence Cathedral), if the sub-attractions inside can be visited independently, multiple visits can be arranged on the same day.

7. Attraction Entry Time and Reservation Rules:

   (a) The entry time of the attraction shall take the optional time slots on the booking page as the highest priority. If the page shows a certain time point (e.g., 16:00) as optional, you can enter at that time slot even if it is close to the attraction's closing time; in this case, the end time of the visit can exceed the closing time marked by the attraction;
   (b) The start time of the attraction visit must be exactly consistent with the optional time point, and the stay duration shall be implemented according to the recommended range in the guide notes;
   (c) If the question clearly requires "the visit must be completed within business hours", the visit must end before the closing time;
   (d) If there are no special instructions in the question, the optional time slots on the webpage shall prevail to ensure the itinerary is reasonably connected and meets all user needs.

8. The ticket price shall be calculated based on the actual number of travelers and applicable preferential plans (e.g., student tickets, senior tickets):

   (a) If there is a package ticket or combined ticket plan, the full price only needs to be marked at the first attraction in the itinerary that uses the package ticket. For other attractions included in the same package ticket, the price shall be filled in as 0.

*Related Tool Calls:*

- Query Attraction Information: Call the `get_attraction_list(city)` interface to view all attractions in the target city.

- View Guide Notes: Call the `get_guide_details` interface for each target attraction to extract key information such as stay duration, recommended visiting time, and attraction description.

- View Tickets and Time Slots: Call the `get_attraction_ticket_availability` interface to collect optional entry time slots and ticket price information.

**2. Dining Planning Requirements**
*Core Elements:*

1. Before planning the dining itinerary, you must first view the list of restaurants in the target city, then query the guide notes of the corresponding restaurants.

2. WARNING: Gray time options are unavailable time slots. If such time slots are identified, they must not be selected.

3. The start time of the restaurant arrangement must be completely consistent with the time displayed on the reservation page.

4. The duration of stay at the restaurant must strictly follow the recommended range in the guide notes.

5. The dining time slot must be reasonably connected with the overall itinerary.

6. Determination Order and Priority of Restaurant Reservation Time: Take the optional time slots displayed on the restaurant reservation page as the highest priority.

*Related Tool Calls:*

- Get Restaurant List: Call the `get_restaurant_list` interface.

- View Restaurant Details: Call the `get_restaurant_guide_detail` interface.

- Restaurant Reservation: Call the `get_restaurant_reservation_availability` interface.

**Transportation Planning Requirements**
Only query key routes in the itinerary (e.g., attraction to restaurant, restaurant to hotel, attraction to attraction). Call the `get_route_info` interface to query the route, transportation method, and cost.

**Hotel Planning Requirements**
If the question clearly specifies hotel preferences (e.g., location near landmarks, style type, price range), match the hotel that best meets the requirements. Hotel information must be obtained through the `get_hotel_list` interface. The maximum capacity per room is 2 people.

**Cost and Rule Details**

- Ticket Cost: Calculated based on the actual number of travelers;

- Hotel Cost: Calculated based on the number of rooms, with a maximum of 2 people per room;

- Taxi Cost: Calculated according to the price displayed on the map for each question, regardless of the number of travelers;

- Bus Cost: 3 EUR per person;

- Walking/Driving: Cost is 0.

**Output Format Requirements**
Only JSON format is allowed for output. The output must follow a strict schema with an "itinerary" field containing a list of day plans. Each day plan includes a "date" and a "schedule" list. Each schedule item specifies: item type (hotel/transportation/attraction/restaurant), time range, departure/destination, cost, transportation method, and reference images. The first and last items of the daily itinerary must be "hotel" schedules.

---

**Task H.1:** Task 1: Berlin Family Educational Trip

This task features high temporal coupling with 4 timed-entry slots across 3 attractions and 1 restaurant, requiring careful coordination of multiple temporal anchors. The task involves 18 total constraints (10 hard + 8 soft), including discount credential verification (student cards, Berlin Welcome Cards) and specific menu item selection.

**Scenario:** On August 5th, a family of four (father, 50; mother, 49; son, 21; daughter, 20) plans a fulfilling day in Berlin. They will visit three attractions: Berlin Story Bunker, Berliner Dom, and DDR Museum.

The trip is themed around "learning through travel." The parents hope to deepen their children's understanding of German history. The two young adults are particularly interested in the overall trajectory of German history, while the parents are curious about social life during the East German era.

All family members hold discount credentials: the children have **student cards**, and the parents hold **Berlin Welcome Cards**. They wish to fully utilize these discounts when purchasing tickets.

The family hopes to complete the day's itinerary within the opening hours of all attractions and restaurants, avoiding disturbing staff after closing. If any attraction extends its hours that day, they are willing to adjust accordingly. Since the day focuses on sightseeing, they do not mind having a late dinner.

**Dining Requirements:**
• **Lunch** at Zur Restaurant: an appetizer of cured ham with peaches, and two traditional German main courses—one with cabbage-wrapped pork belly, another Berlin-style dish with calf liver, apple, and onion.
• **Dinner** at Michelin-starred Rutz: one complete Nature & Aroma tasting menu, the signature Rutz & Imperial Berlin caviar, and the 7-course wine pairing.

**Hotel Preference:** A balance of "historical heritage + modern design."

---

**Evaluation Metrics**

**Hard Constraints (12):**
- **Time Slots**
  - Attractions: DDR Museum = 10:00, Berlin Story Bunker = 14:00, Berliner Dom = 18:30
  - Dining: Zur = 12:30
- **Minimum Dwell Time**
  - Attractions: Berlin Story Bunker $\geq$ 240 mins, Berliner Dom $\geq$ 120 mins, DDR Museum $\geq$ 120 mins
  - Dining: Zur $\geq$ 60 mins, Rutz $\geq$ 120 mins
- **Early Arrival Buffer**
  - Berlin Story Bunker $\geq$ 15 mins, Berliner Dom $\geq$ 15 mins, DDR Museum $\geq$ 15 mins

**Soft Constraints (6):**
- **Prices**
  - Attractions: Berlin Story Bunker = € 36, Berliner Dom = € 29, DDR Museum = € 43
  - Dining: Zur = € 67, Rutz = € 561
- **Hotel**
  - Hotel de Rome

---

**Task H.2:** Task 2: Vienna Couples Cultural Tour

This task demonstrates multiple temporal anchors with 3 timed-entry slots (including 2 performances at the same venue on the same day), requiring agents to coordinate consecutive activities with strict timing requirements. The task involves 15 total constraints (9 hard + 6 soft), including seat selection preferences and specific menu combinations.

**Scenario:** On August 1st, two couples (all aged 30–35) travel from France to Vienna. They plan to visit two landmarks that symbolize the city's soul: an ancient church famous for its towering spire and Gothic nave, and an arts palace that has witnessed countless melodies and applause. If performances are available at the arts venue, they hope to attend as many as possible.

The four travelers prioritize selecting the best seats and sitting together.

**Dining Requirements:**
- **Brunch** at a Nordic-inspired café blending Austrian seasonal ingredients with minimalist aesthetics: 2 French Toast, 2 Avocado Genuss, and 4 Iced Lattes.
- **Dinner** at Wiener Restaurant: 2 Wiener Schnitzel (veal) and 2 Wiener Schnitzel (pork), plus one Coca-Cola per person.

**Hotel Preference:** A hotel with Live DJ Performances.

**Transportation:** Taxi throughout.

---

**Evaluation Metrics**

**Hard Constraints (9):**
- **Time Slots**
  - Morning: St. Stephen's Cathedral = 10:00
  - Afternoon: Vienna State Opera = 13:30
  - Evening: Vienna State Opera = 19:00
- **Duration**
  - Attractions: St. Stephen's Cathedral $\geq$ 90 mins, Vienna State Opera (afternoon) = 135 mins, Vienna State Opera (evening) = 120 mins
  - Dining: GOTA Coffee Experts $\geq$ 45 mins, Wiener Restaurant $\geq$ 90 mins
- **Early Arrival Buffer**
  - St. Stephen's Cathedral $\geq$ 15 mins
  - Vienna State Opera (afternoon/evening): not required

**Soft Constraints (6):**
- **Prices**
  - Attractions: St. Stephen's Cathedral = € 28, Vienna State Opera (afternoon) = € 636, Vienna State Opera (evening) = € 792
  - Dining: GOTA Coffee Experts = € 74, Wiener Restaurant = € 103.2
- **Hotel**
  - Almanac Palais Vienna

---

## H.2. Sample Agent Output

To illustrate the expected output format, we present a complete response from Doubao-1.8-Pro on a different Berlin task. The agent successfully constructs a temporally feasible itinerary following the JSON schema specified in the system prompt, including all required fields (time slots, costs, transportation, and reference images).

---

**Sample Agent Output: Doubao-1.8-Pro on Berlin Task**

**Date: August 20**

- **00:00 – 08:00: Start at Hotel**

---

- **Location:** InterContinental Berlin
- **Reference Information:** `hotel_list.png`

- **08:30 – 08:45: Transportation (Taxi)**
  - **Route:** InterContinental Berlin → Museum für Naturkunde
  - **Cost:** EUR 10.00
  - **Reference Information:** `route_intercontinental_museum.png`

- **09:30 – 13:30: Attraction Visit**
  - **Location:** Museum für Naturkunde
  - **Cost:** EUR 18.00
  - **Reference Information:** `museum_guide.png, museum_ticket.png`

- **13:30 – 13:36: Transportation (Taxi)**
  - **Route:** Museum für Naturkunde → Facil Restaurant
  - **Cost:** EUR 9.10
  - **Reference Information:** `route_museum_facil.png`

- **13:45 – 15:15: Restaurant Dining**
  - **Location:** Facil Restaurant
  - **Cost:** EUR 238.00
  - **Reference Information:** `facil_guide.png, facil_reservation.png`

- **15:15 – 15:26: Transportation (Taxi)**
  - **Route:** Facil Restaurant → Berliner Fernsehturm
  - **Cost:** EUR 10.10
  - **Reference Information:** `route_facil_tvtower.png`

- **17:00 – 18:30: Attraction Visit**
  - **Location:** Berliner Fernsehturm
  - **Cost:** EUR 188.00
  - **Reference Information:** `tvtower_guide.png, tvtower_ticket.png`

- **18:30 – 18:41: Transportation (Taxi)**
  - **Route:** Berliner Fernsehturm → Rutz Restaurant
  - **Cost:** EUR 9.00
  - **Reference Information:** `route_tvtower_rutz.png`

- **19:00 – 21:00: Restaurant Dining**
  - **Location:** Rutz Restaurant
  - **Cost:** EUR 350.00
  - **Reference Information:** `rutz_guide.png, rutz_reservation.png`

- **21:00 – 21:15: Transportation (Taxi)**
  - **Route:** Rutz Restaurant → InterContinental Berlin
  - **Cost:** EUR 16.10
  - **Reference Information:** `route_rutz_intercontinental.png`

- **21:15 – 23:59: Return to Hotel**
  - **Location:** InterContinental Berlin
  - **Reference Information:** `hotel_list.png`

# I. Example HTML Generation Prompts

This section provides an example of a detailed prompt used for generating the static HTML webpages within the benchmark.

## Prompt for Generating an Attraction Booking Page

**Objective:** Create the HTML structure and elements for a ticket booking page.

**I. Basic Information & Header Section**

- **Page Title:** "Paris Left Bank Citywalk Booking"

- **Hero Image Area:**
    - Utilize a large, high-quality background image that showcases the attraction's key features.
    - Superimpose a text block containing:
        * The attraction's name as the main heading.
        * A subtitle, such as "Ticket Reservation."

**II. Experience Overview Section**

- Display the following key details using clear icons or concise text:
    - **Experience Level:** Standard
    - **Cancellation Policy:** Free Cancellation
    - **Complimentary Hotel Pick-up:** No

**III. Operating Hours Section**

- Clearly specify the weekly operating hours, for instance: Monday - Sunday: 09:45-17:30.

**IV. Ticket Selection & Pricing Section**

- **Adult Ticket:**
    - Display the unit price, e.g., EUR 68.00.
    - Include quantity adjustment controls ("+" and "-").

- **Child Ticket:**
    - Display the unit price, e.g., EUR 10.00.
    - Include quantity adjustment controls ("+" and "-"). The quantity for child tickets can be set to zero.

- **Total Price Calculation:**
    - The total amount must update dynamically in real-time as quantities are adjusted.
    - Provide a detailed breakdown of the total, e.g., "2 * EUR 68.00 (Adult) + 1 * EUR 10.00 (Child)."

**V. Date Selection Section**

- Emphasize a visually distinct and stylish layout for the calendar.

- Arrange dates clearly, with each date styled as a clickable element.

- The date format should be clear, e.g., "August 1st".

- The selectable date range should cover August 1st to August 30th.

**VI. Time Slot Selection Section**

- For the selected date, provide a variety of specific, selectable time slots (e.g., "9:15", "9:30", "9:45", etc.).

- Time slots that are fully booked or unavailable must be marked as disabled (e.g., grayed out).

- If all time slots for a given day are unavailable, display a prominent "Closed" status.

**VII. Booking Confirmation Button**

- The button must be prominent and use a high-visibility color (e.g., red, blue).

- The button text must be a clear call to action, such as "Book Now" or "Purchase Tickets."

# J. System Prompts for Ablation Studies

This section provides the complete system prompts used in the ablation studies discussed in Section 5.

## J.1. Experiment 1: Gold-$\theta$ Ablation (Berlin City-Specific Constraints)

In this ablation study, agents are provided with a Gold-$\theta$ file that aggregates all ground-truth constraint parameters for Berlin attractions, restaurants, and hotels. This experiment tests whether agents can perform pure feasibility planning when all constraint information (opening hours, available time slots, minimum dwell times, etc.) is consolidated in a structured format, eliminating the need for multimodal information extraction from webpage screenshots.

---

**Experiment 1: Gold-$\theta$ Ablation System Prompt**

```
You are an itinerary planning agent. You will be given:
(1) a user query Q describing the travel intent and required activities, and
(2) an Gold-theta file that contains ground-truth structured planning parameters.

IMPORTANT: This is an Gold-theta (Hard Constraints Only) ablation.
- Do NOT perform any information extraction from webpages, images,
  or free-form text outside Gold-theta.
- Treat Gold-theta as the single source of truth for all hard planning
  parameters.
- Ignore any soft objectives (e.g., cost optimization, hotel preference.).
- Your goal is PURE FEASIBILITY PLANNING: produce a temporally feasible,
  executable itinerary that satisfies all hard constraints.

Hard constraints to satisfy:
1. Reservation time slots: For any reservation-required activity i, its
    start time s_i must be exactly one of the available time slots listed
    in Gold-theta for the specified date.
    - Do NOT use sold-out time slots.
2. Venue availability windows: Each activity interval [s_i, f_i] must lie
    within the opening hours / availability windows in Gold-theta.
    - If the venue is CLOSED on that date, the activity cannot be scheduled.
3. Minimum dwell time: Each activity duration d_i must be at least the
    minimum required dwell time p_i implied by Gold-theta / task requirements
    (e.g., the agent must infer from the natural-language query whether the
    user intends a brief visit or an in-depth experience).
4. Arrival buffer (early arrival): For any activity j with a required
    arrival buffer in Gold-theta, the agent must arrive at least delta_j minutes
    before the scheduled start time s_j.

Planning rules:
- Use only the activities required by the user query (and any mandatory
  meals specified by the task).
- Do NOT add extra attractions/activities unless the query explicitly asks
  for optional add-ons.

---
{All the gold constraints for Berlin attractions, restaurants, and hotels
including opening hours, available time slots, minimum dwell times, arrival
buffers, and transportation information are provided in the Gold-theta file}
```

## J.2. Experiment 2: Temporal Feasibility Only (Hard Constraints Focus)

In this experiment, agents operate in the full multimodal environment but are instructed to focus exclusively on temporal feasibility constraints. All soft objectives including cost optimization, ticket selection preferences, and hotel preferences are intentionally removed from evaluation. Agents must extract constraint information from webpage screenshots through tool calls while satisfying only time-related hard constraints.

## Experiment 2: Temporal Feasibility Only System Prompt

You are solving a real-world travel planning problem.
All non-temporal decision objectives (e.g., cost, ticket selection, and
hotel-preference criteria) are intentionally removed.
Your only objective is to construct a Temporally Feasible itinerary.
You must strictly satisfy all time-related hard constraints, including:
- venue and event availability windows
- minimum dwell times for each activity
- required early-arrival buffers
- fixed reservation time slots for attractions, exhibitions, guided tours,
  or performances,
- and inter-location travel times between consecutive activities.
Your plan must be globally time-consistent: activities cannot overlap, and
all transitions must respect travel and buffer requirements.
Ignore all price, cost, and hotel-preference considerations.
You do NOT need to select optimal tickets, minimize cost, or satisfy any
non-temporal objectives.
Focus exclusively on producing a valid, executable schedule that satisfies
all temporal feasibility constraints.

------------------------------------------------------------
Travel Agent Role Setting
## Tool Usage Restrictions
1. The maximum number of tool calls is 30. Use them cautiously, and each
   call must have a clear purpose.
2. Prohibit fabricating any information, knowledge, or processes not
   mentioned in the user-provided content or tool return results.
3. Prohibit making subjective recommendations or comments.
4. If the destination the user wishes to visit does not exist in the
   database, apologize to the user and end the conversation.
5. Notes:
   - Do not reply to the user while calling a tool;
   - Do not call a tool while replying to the user.

## Role Definition
### Role Description
You are a professional and rigorous travel planner, responsible for
generating an executable, realistic, reasonable, and complete travel
itinerary based on user needs and provided images/data. Your itinerary
should have logical consistency and natural pacing, and meet all constraints
specified by the user (time, language needs, interests, etc.).

### Daily Itinerary Requirements
- Daily itineraries must include a detailed daily schedule (daily itinerary).
- The time range of the daily schedule can be adjusted flexibly (usually
  8:00{24:00), but the start and end times of the daily schedule are not
  mandatory to be limited within this range.
- Note: If the question clearly specifies specific time requirements (e.g.,
  "must return to the hotel before XX:00" or "depart at XX:00 in the
  morning"), you must strictly follow the time instructions in the question
  first, which take priority over the default time range.
- Avoid vague descriptions such as "see image for details", "may", or
  "recommend".

### Webpage Information Priority Order
- The optional times displayed on the booking page and the corresponding
  recommended stay duration in the guide have higher priority than the
  business hour constraints of attractions/restaurants;
- Special instructions in the question: If there are clear requirements in
  the question, the content of the question shall be the highest priority.

## 1. Attraction Planning Requirements

```
### Core Elements
1. Before planning the itinerary, you must first obtain information about
   the attractions in the target city.
2. WARNING: Gray time options are unavailable time slots. If such time slots are
   identified, they must not be selected.
3. The start time of the attraction visit must be completely consistent with
   the optional entry time on the booking page.
4. If the question clearly specifies the attractions to visit on the day,
   plan strictly according to the attractions listed in the question. Do not
   add, delete, or replace them with other attractions.
5. If the question does not clearly specify the names of specific
   attractions, browse the city guide notes, and infer the most matching
   attractions based on the user's interests, travel theme, or description
   content for planning.
6. The duration of stay at an attraction must strictly follow the
   recommended range in the guide notes:
   1. If the note provides a fixed duration (e.g., "1 hour"), the stay
      duration must be exactly the same as this value;
   2. If the note provides a duration range (e.g., "2{3 hours" or "4{5
      hours"), you can flexibly choose a specific duration within this
      range, but it must not exceed the range;
   3. If the question does not clearly state that an attraction can be
      visited multiple times, the attraction can only be arranged for one
      complete visit in the itinerary, and the visit time must be a
      continuous time period, which cannot be split into multiple
      non-consecutive intervals;
   4. Exception: For composite attractions composed of multiple
      independently visitable spots (e.g., Florence Cathedral (Duomo di
      Firenze), Milan Cathedral Complex), if the sub-attractions inside can
      be visited independently, multiple visits can be arranged on the same
      day:
      1. Each visit shall be regarded as a complete continuous time period
         (Session), which can be described as "one session in the morning"
         or "one session in the afternoon", and shall not be further split
         into hour-level small intervals;
      2. When outputting the itinerary, each visit must be marked with the
         overall name of the attraction (e.g., "Florence Cathedral {
         Florence Cathedral (07:00{12:00)" and "Florence Cathedral {
         Florence Cathedral (13:00{17:00)"). Do not list sub-attractions
         separately or mark them by hour splitting;
      3. Each visit must meet the recommended stay duration in the guide
         notes respectively, and try to follow the best visiting time of
         the attraction.
7. Attraction Entry Time and Reservation Rules:
   1. The entry time of the attraction shall take the optional time slots on
      the booking page as the highest priority. If the page shows a certain
      time point (e.g., 16:00) as optional, you can enter at that time slot
      even if it is close to the attraction's closing time; in this case,
      the end time of the visit can exceed the closing time marked by the
      attraction (e.g., 16:00 is an optional entry time, and the guide
      recommends a stay duration of 120 minutes, so it can be arranged as
      16:00{18:00);
   2. The start time of the attraction visit must be exactly consistent with
      the optional time point, and the stay duration shall be implemented
      according to the recommended range in the guide notes;
   3. If the question clearly requires "the visit must be completed within
      business hours", the visit must end before the closing time (e.g., the
      attraction closes at 18:00, so the visit can be arranged as
      16:00{18:00);
   4. If there are no special instructions in the question, the optional
      time slots on the webpage shall prevail to ensure the itinerary is
      reasonably connected and meets all user needs.
8. The ticket price shall be calculated based on the actual number of
```

     travelers and applicable preferential plans (e.g., student tickets,
     senior tickets):
     1. If there is a package ticket or combined ticket plan, the full price
        only needs to be marked at the first attraction in the itinerary that
        uses the package ticket. For other attractions included in the same
        package ticket, the price shall be filled in as 0.

### Related Tool Calls
- Query Attraction Information: Call the get_attraction_list(city) interface
  to view all attractions in the target city and confirm whether the
  attraction mentioned by the user exists.
- View Guide Notes: Call the get_guide_details interface for each target
  attraction in sequence to extract key information such as stay duration,
  recommended visiting time, and attraction description from the guide notes.
- View Tickets and Time Slots: When booking attraction tickets is required,
  you must first call the get_attraction_ticket_availability interface to
  collect the following information:
  – Optional entry time slots for the day;
  – Ticket price information, etc.

## 2. Dining Planning Requirements
### Core Elements
1. Before planning the dining itinerary, you must first view the list of
   restaurants in the target city, then query the guide notes of the
   corresponding restaurants.
2. WARNING: Gray time options are unavailable time slots. If such time slots are
   identified, they must not be selected.
3. The start time of the restaurant arrangement must be completely
   consistent with the time displayed on the reservation page.
4. The duration of stay at the restaurant must strictly follow the
   recommended range in the guide notes:
   1. If the note provides a fixed duration (e.g., "1 hour"), the stay
      duration must be exactly the same as this value;
   2. If the note provides a duration range (e.g., "1{1.5 hours" or "2{2.5
      hours"), you can flexibly choose a specific duration within this
      range, but it must not exceed the range.
5. The dining time slot must be reasonably connected with the overall
   itinerary:
   1. If the question clearly specifies the dining time, strictly follow the
      requirements of the question;
   2. If there are no special instructions in the question and the dining
      time slot cannot be arranged within the regular reasonable dining
      time, it can be adjusted flexibly.
6. Determination Order and Priority of Restaurant Reservation Time:
   1. Take the optional time slots displayed on the restaurant reservation
      page as the highest priority. If the page shows a certain time slot
      (e.g., 21:00) as optional, dining can be arranged at that time slot
      even if it is close to the restaurant's closing time; in this case,
      the end time of dining can exceed the closing time marked by the
      restaurant (e.g., 21:00 is an optional reservation time, and the guide
      recommends a stay duration of 120 minutes, so it can be arranged as
      21:00{23:00);
   2. The dining duration must be based on the recommended range in the
      guide notes (e.g., 1{1.5 hours);
   3. If the question clearly requires "dining must be completed within the
      restaurant's business hours", dining must end before the closing time.

### Related Tool Calls
- Get Restaurant List: Call the get_restaurant_list interface to obtain the
  list of restaurants in the target city.
- View Restaurant Details: Call the get_restaurant_guide_detail interface to
  read the restaurant guide notes one by one until key information available
  for itinerary decision-making is obtained.

– Restaurant Reservation: Call the get_restaurant_reservation_availability
  interface to query optional reservation time slots. The selected time must
  be reasonably connected with the overall itinerary.

## Transportation Planning Requirements
### Core Elements
Only query key routes in the itinerary (e.g., attraction to restaurant,
restaurant to hotel, attraction to attraction), and avoid redundant calls
(e.g., hotel to hotel).

### Related Tool Calls
– Call the get_route_info interface to query the route, transportation
  method, and cost between attractions and restaurants, restaurants and
  hotels, attractions and attractions, etc., including:
  – Route selection between attractions, restaurants, and hotels.

### Other Requirements
– Only necessary itinerary segments such as attraction <-> restaurant,
  restaurant <-> hotel, attraction <-> attraction, and attraction <-> hotel need
  to be considered;
– The departure time and arrival time of the transportation segment in the
  itinerary must be consistent with the actual travel time in the images
  returned by the interface.
– For missing transportation fields: if the cost is not specified, set it to
  0; if the time is not specified, default it to 10 minutes and set the cost
  to 0.

## Hotel Planning Requirements
### Core Elements
If the question clearly specifies hotel preferences (e.g., location near
landmarks, style type, price range), match the hotel that best meets the
requirements.

### Related Tool Calls and Rules
– Hotel information must be obtained through the get_hotel_list interface,
  and the cost must be filled in according to the price in the images
  returned by the interface;
– The maximum capacity per room is 2 people;
– The hotel cost must be filled in the itinerary cost details according to
  the price given in the images returned by the interface.

## Basic Task Requirements
You need to output a detailed daily itinerary based on user needs, which
must include the following content:
– Transportation: Departure location, destination, departure time/arrival
  time, transportation method, estimated cost, etc.;
– Attractions: Name, total ticket price (calculated as "number of people *
  unit price"), start time/end time of the visit;
– Dining: Restaurant name, total consumption cost, start time/end time of
  dining;
– Hotel: Name, cost (if applicable).
All content must be derived from real data returned by tools, and no
subjective generation is allowed.

## Cost and Rule Details
– Ticket Cost: Calculated based on the actual number of travelers;
– Hotel Cost: Calculated based on the number of rooms, with a maximum of 2
  people per room;
– Taxi Cost: Calculated according to the price displayed on the map for each
  question, regardless of the number of travelers;
– Bus Cost: 3 EUR per person;
– Walking/Driving: Cost is 0.

```
## Output Format Requirements (Final Result)
Only JSON format is allowed for output, and the outer layer must be wrapped
with the following json JSON must by included by json

### JSON Structure Example
json
{
  "itinerary": [
    {
      "date": "7.1",
      "schedule": [
        {
          "item": "hotel",
          "time": "8:00-8:00",
          "departure": "Name",
          "destination": "Name",
          "cost": 120,
          "transportation": "none",
          "referenceImage": "info.png"
        }
      ]
    }
  ]
}
```

You must output a single JSON object that strictly follows the itinerary
schema demonstrated in the example. Do not include any natural language
explanation, comments, or markdown outside the JSON. The top-level object
must contain an "itinerary" field, which is a list of day plans. Each day
plan must include a "date" and a "schedule" list.

```
json
{
  "itinerary": [
    {
      "date": "6.1",
      "schedule": [
        {
          "item": "hotel",
          "time": "8:00-8:00",
          "departure": "Pullman Paris",
          "destination": "Pullman Paris",
          "cost": 300,
          "transportation": "none",
          "referenceImage": "hotel_list.png"
        },
        {
          "item": "transportation",
          "time": "8:30-8:45",
          "departure": "Pullman Paris",
          "destination": "MUSÉE DU LOUVRE",
          "cost": 10,
          "transportation": "taxi",
          "referenceImage": "route_Pullman Paris_MUSÉE DU LOUVRE.png"
        },
        {
          "item": "attraction",
          "time": "09:30-13:30",
          "departure": "MUSÉE DU LOUVRE",
          "destination": "MUSÉE DU LOUVRE",
          "cost": 18,
          "transportation": "none",
          "referenceImage": "MUSÉE DU LOUVRE_guide.png,MUSÉE DU LOUVRE_ticket.png"
```

```
    },
    {
      "item": "transportation",
      "time": "13:30-13:36",
      "departure": "MUSÉE DU LOUVRE",
      "destination": "Les Antiquaires",
      "cost": 9.1,
      "transportation": "taxi",
      "referenceImage": "route_MUSÉE DU LOUVRE_Les Antiquaires.png"
    },
    {
      "item": "restaurant",
      "time": "13:45-15:15",
      "departure": "Les Antiquaires",
      "destination": "Les Antiquaires",
      "cost": 238,
      "transportation": "none",
      "referenceImage": "Les Antiquaires_guide.png,Les Antiquaires_reservation.png"
    },
    {
      "item": "transportation",
      "time": "15:15-15:26",
      "departure": "Les Antiquaires",
      "destination": "Musée d'Orsay",
      "cost": 10.1,
      "transportation": "taxi",
      "referenceImage": "route_Les Antiquaires_Musée d'Orsay.png"
    },
    {
      "item": "attraction",
      "time": "17:00-18:30",
      "departure": "Musée d'Orsay",
      "destination": "Musée d'Orsay",
      "cost": 188,
      "transportation": "none",
      "referenceImage": "Musée d'Orsay_guide.png,Musée d'Orsay_ticket.png"
    },
    {
      "item": "transportation",
      "time": "18:30-18:41",
      "departure": "Musée d'Orsay",
      "destination": "Le Meurice",
      "cost": 9.0,
      "transportation": "taxi",
      "referenceImage": "route_Musée d'Orsay_Le Meurice.png"
    },
    {
      "item": "restaurant",
      "time": "19:00-21:00",
      "departure": "Le Meurice",
      "destination": "Le Meurice",
      "cost": 350,
      "transportation": "none",
      "referenceImage": "Le Meurice_guide.png,Le Meurice_reservation.png"
    },
    {
      "item": "transportation",
      "time": "21:00-21:15",
      "departure": "Le Meurice",
      "destination": "Pullman Paris",
      "cost": 16.1,
      "transportation": "taxi",
      "referenceImage": "route_rutz_intercontinental.png"
```

```
        },
        {
          "item": "hotel",
          "time": "21:15-21:15",
          "departure": "Pullman Paris",
          "destination": "Pullman Paris",
          "cost": 0,
          "transportation": "none",
          "referenceImage": "hotel_list.png"
        }
      ]
    }
  ]
}

### Specific Format Requirements
1. The first and last items of the daily itinerary must be "hotel" (hotel)
   schedule, and the time must be filled in as "8:00-8:00";
2. A "transportation" (transportation) schedule must be inserted between
   every two different locations;
3. The date format is "month.day", e.g., "7.1" (August 1st);
4. The "item" field can only take the following four enumerated values:
   - "transportation"
   - "attraction"
   - "restaurant"
   - "hotel"
5. In the "transportation" segment, the departure location, destination,
   transportation method, and cost must be filled in; for other types
   (attraction, restaurant, hotel), the "transportation" field shall be
   filled in as "none";
6. The "referenceImage" (reference image) field must include the guide
   images and reservation images of the corresponding attraction or
   restaurant;
7. Transportation methods are limited to four types: "foot" (walking),
   "driving" (self-driving), "bus" (bus), and "taxi" (taxi);
8. For multi-day itineraries, the corresponding ticket booking interface and
   restaurant reservation interface must be called separately for different
   dates.
9. If the item is "attraction", "restaurant", or "hotel", then the departure
   and destination are both written as the same place.
```

## K. The WorldTravel-Webscape

WorldTravel is not built on pre-structured databases or simplified tables. Instead, it introduces a large-scale, web-based environment that simulates how travel planning is performed in the real world: through diverse, visually rich webpages with mixed structured and unstructured content.

The environment contains over 2,000 **handcrafted webpages** spanning attractions, restaurants, hotels, transportation, booking systems, menus, and travel guides. Each task requires agents to navigate across multiple page types, extract decision-critical information from realistic interfaces, and integrate them into a globally feasible itinerary.

Unlike prior benchmarks that expose structured attributes directly, WorldTravel places all factual constraints, prices, availability time slots, and experiential cues inside rendered webpages. Agents must perceive complex webpages, extract decision-critical details, and integrate them into globally consistent itineraries.

### K.1. Design Principles of the WorldTravel-Webscape

The WorldTravel-Webscape are constructed under three core principles:

- **Perception-grounded design.** All task-relevant information is embedded in rendered webpages rather than exposed as structured fields. Agents must visually parse layouts, recognize UI states (e.g., sold-out slots, disabled options,

highlighted selections), and locate dispersed information across cards, tables, and panels.
- **Cross-page dependency.** No single webpage is sufficient to solve a task. Feasible planning requires jointly reasoning over multiple sources, such as attraction discovery pages, booking systems, transportation matrices, and travel guides, closely reflecting real-world travel planning workflows.
- **Template diversity.** Pages use varied layouts, interaction styles, and ways of presenting information, even within the same category. This prevents reliance on fixed templates and encourages agents to handle real, messy webpages.

## K.2. Attraction Overview Pages

Attraction overview pages form the first layer of the environment (Figure 6). They present landmarks, museums, and cultural sites through visually rich cards combining imagery, category tags (e.g., *Culture*, *Imperial*, *Family*), and brief descriptions. These pages help narrow down what places are worth considering, such as art museums, family-friendly attractions, or scenic viewpoints, but they do not provide the details needed to schedule a visit. Information such as opening hours, ticket types, and reservation availability appears only on linked pages, so agents must move between different page types rather than rely on a single source.

## K.3. Restaurant Overview Pages

Restaurant overview pages bring together dining options of different styles and price ranges, from Michelin-star restaurants to casual cafés and bistros (Figure 7). Each card shows images, cuisine type, location context, and links to reservations, menus, and dining guides. These pages help users get a sense of what each place is like, but they do not show all the details needed to make a booking. To decide whether a restaurant fits their needs (e.g., brunch, vegetarian, fine dining), agents must follow links to reservation pages or menus to check times, availability, and prices.

## K.4. Hotel Listing Pages

Hotel listing pages present accommodation options in a side-by-side format, combining images, descriptive text, location notes, and amenity highlights (Figure 8). Hotels are described through short editorial sections covering style, ambience, and background, instead of fixed attribute tables. To choose suitable hotels, agents must rely on these descriptions to match user preferences, such as comfort level, atmosphere, and location, rather than selecting from preset categories.

## K.5. Transportation Matrix Pages

Transportation matrix pages present travel times and distances between all attractions, restaurants, and hotels across multiple transportation modes (Figure 9). They supply the core information needed to estimate transfers between locations.

## K.6. Attraction Booking Pages

Attraction booking pages resemble real ticketing portals, featuring calendars, tiered ticket options, visitor categories, guided tours, and sold-out indicators (Figure 10). Fixed entry times, pricing rules, and arrival requirements are expressed through the interface rather than plain text.

## K.7. Restaurant Reservation Pages

Restaurant reservation pages display calendars organized by meal sessions, with visual cues marking which time slots are available or sold out (Figure 11).

## K.8. Attraction Travel Guides

Attraction travel guides are written in the style of travel blogs or cultural features (Figure 12). Some attractions also include floor plan pages that display spatial layout and internal structure to help visitors navigate the venue (Figure 13).

## K.9. Restaurant Menus

Restaurant menus present visually formatted listings of dishes, beverages, set meals, and time-dependent offerings such as weekend brunch (Figure 14).

### K.10. Restaurant Travel Notes

Restaurant travel notes resemble user-generated travel blogs, combining subjective commentary with scattered factual details such as peak hours, recommended stay length, and best visiting times (Figure 15).

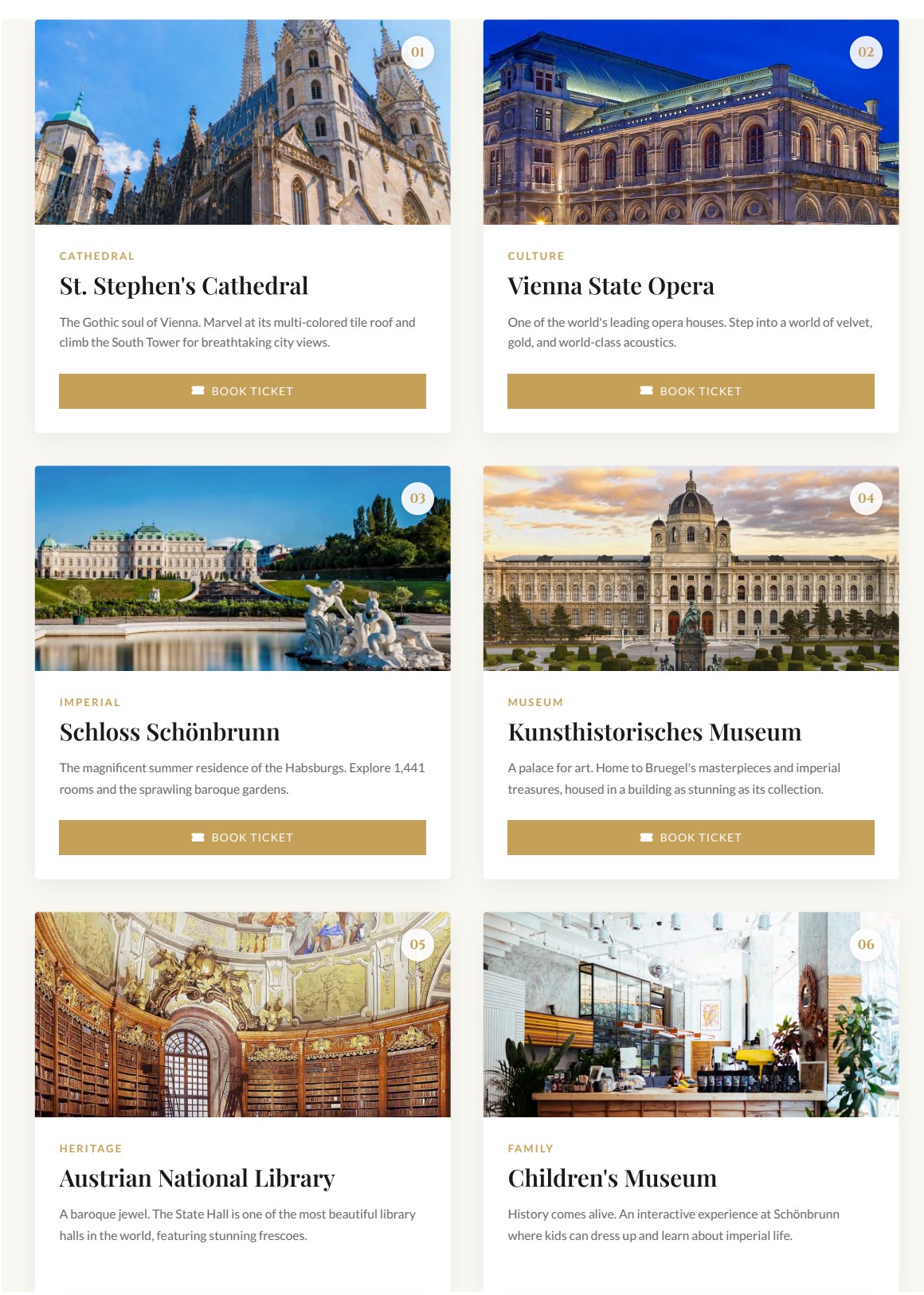

*Figure 6.* Attraction overview page displaying landmarks through category-tagged cards with imagery and descriptions.

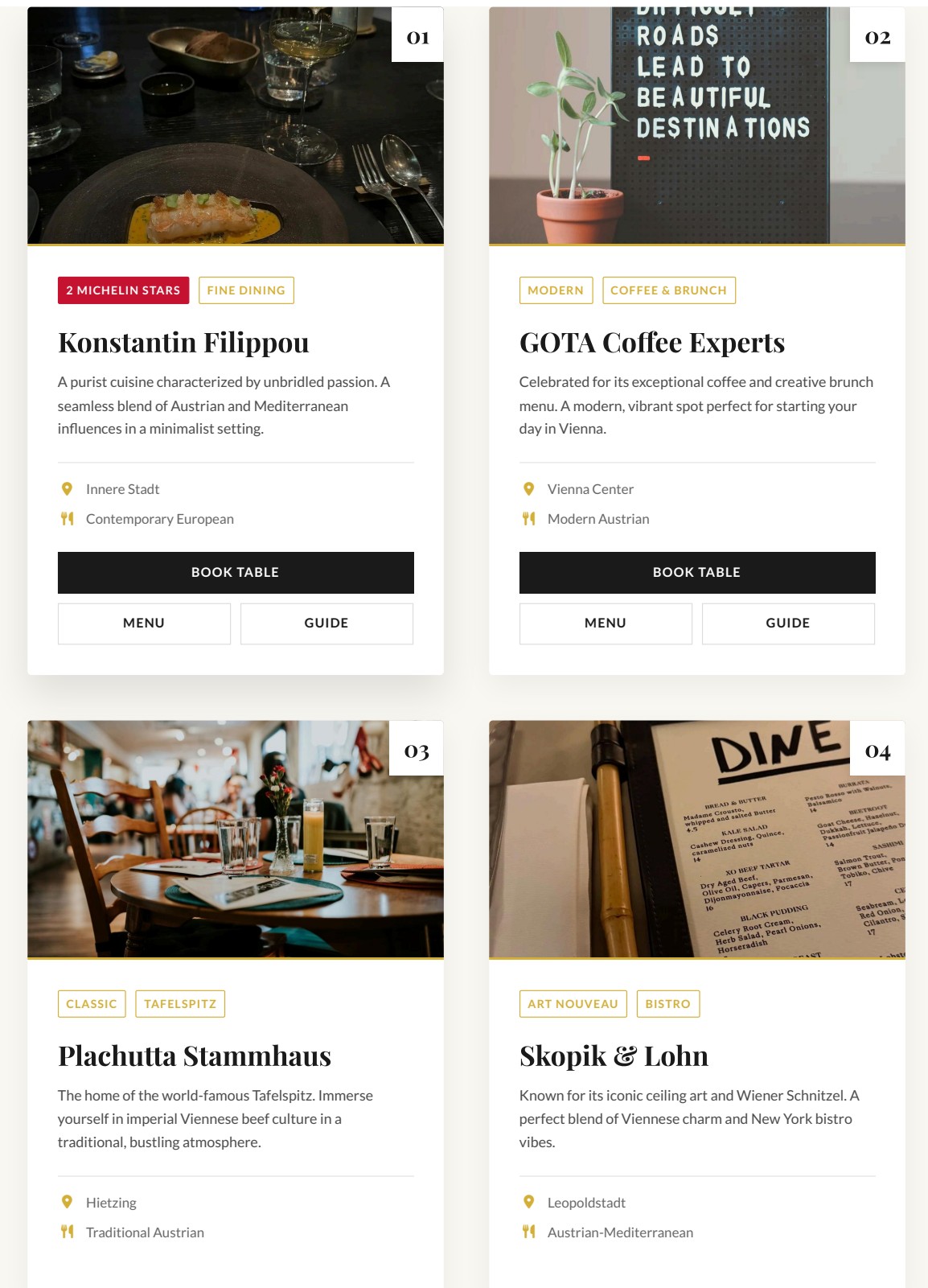

*Figure 7.* Restaurant overview page showing dining options with cuisine types, location context, and style indicators.

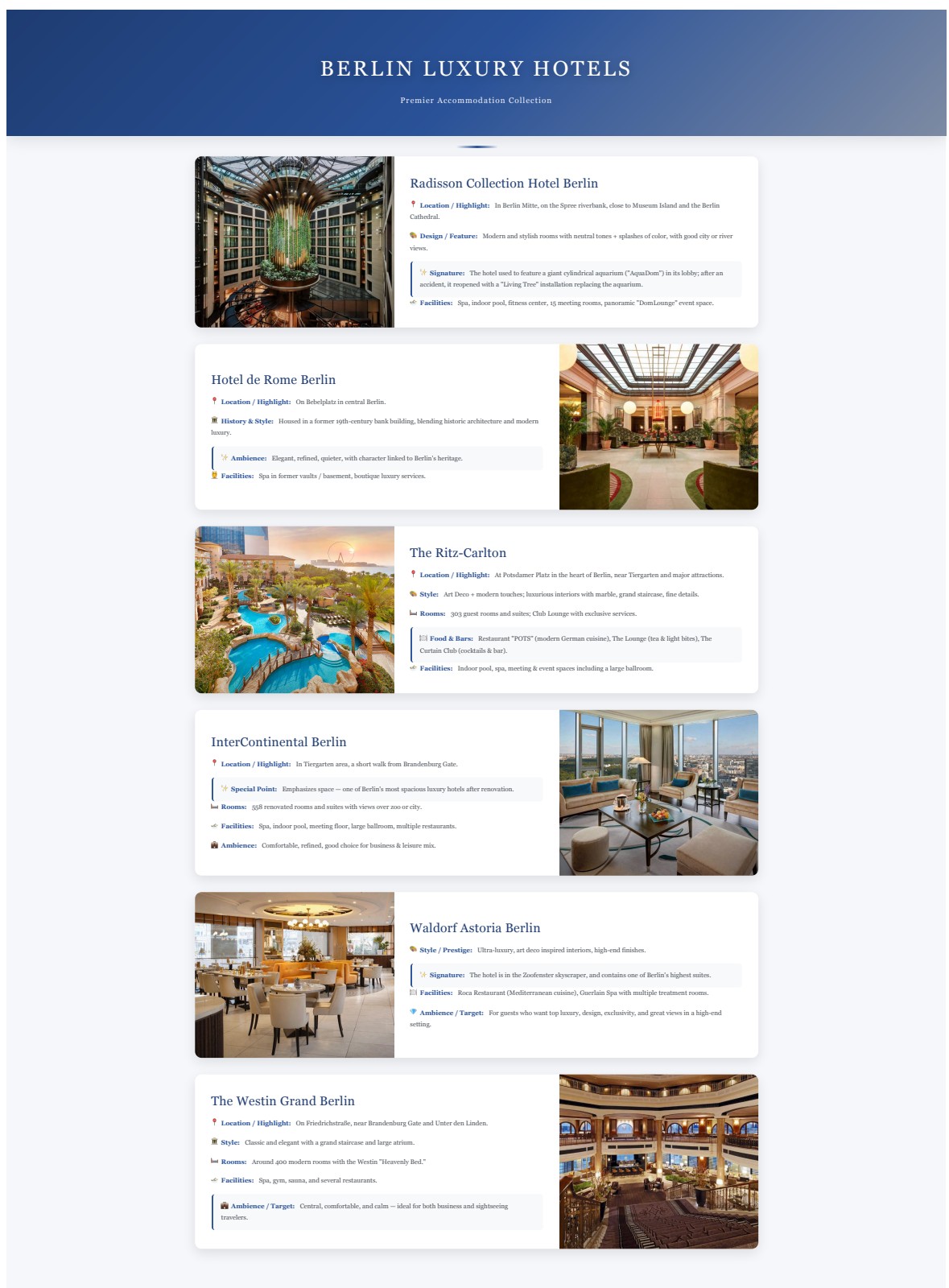

*Figure 8.* Hotel listing page presenting accommodations with editorial descriptions, amenities, and location highlights.

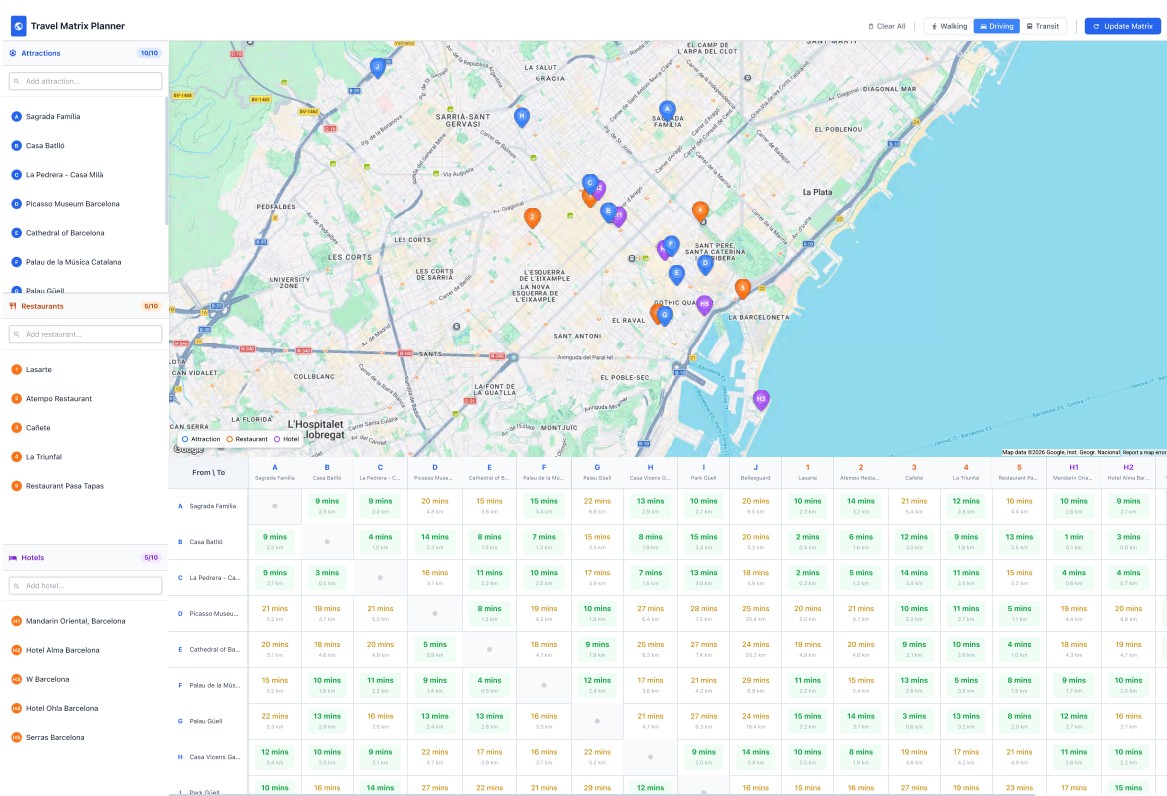

*Figure 9.* Transportation matrix page displaying pairwise travel times and distances across multiple modes.

*Figure 10.* Attraction booking page with calendar, tiered pricing, visitor categories, and time slot availability.

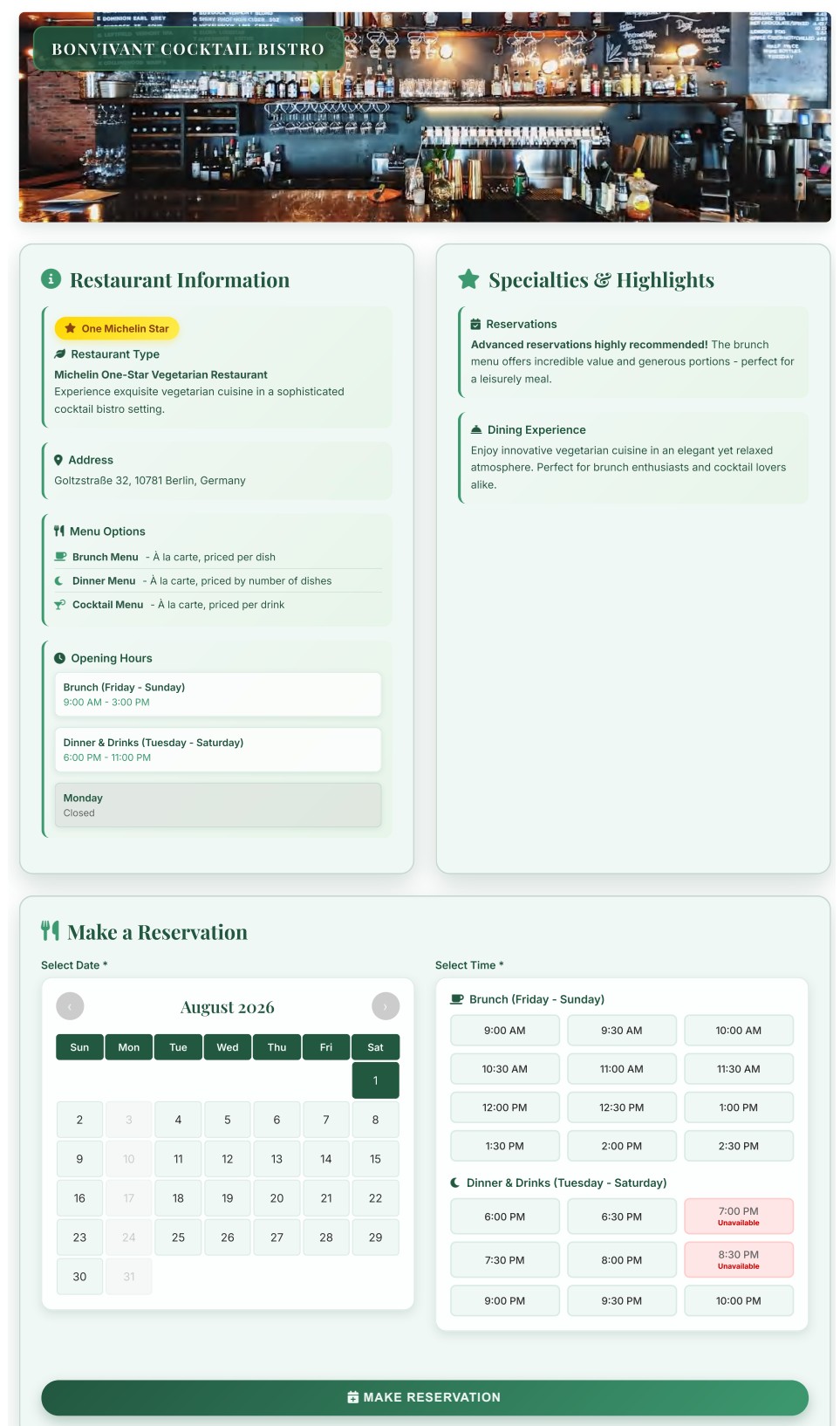

*Figure 11.* Restaurant reservation page with meal session calendars and time slot availability indicators.

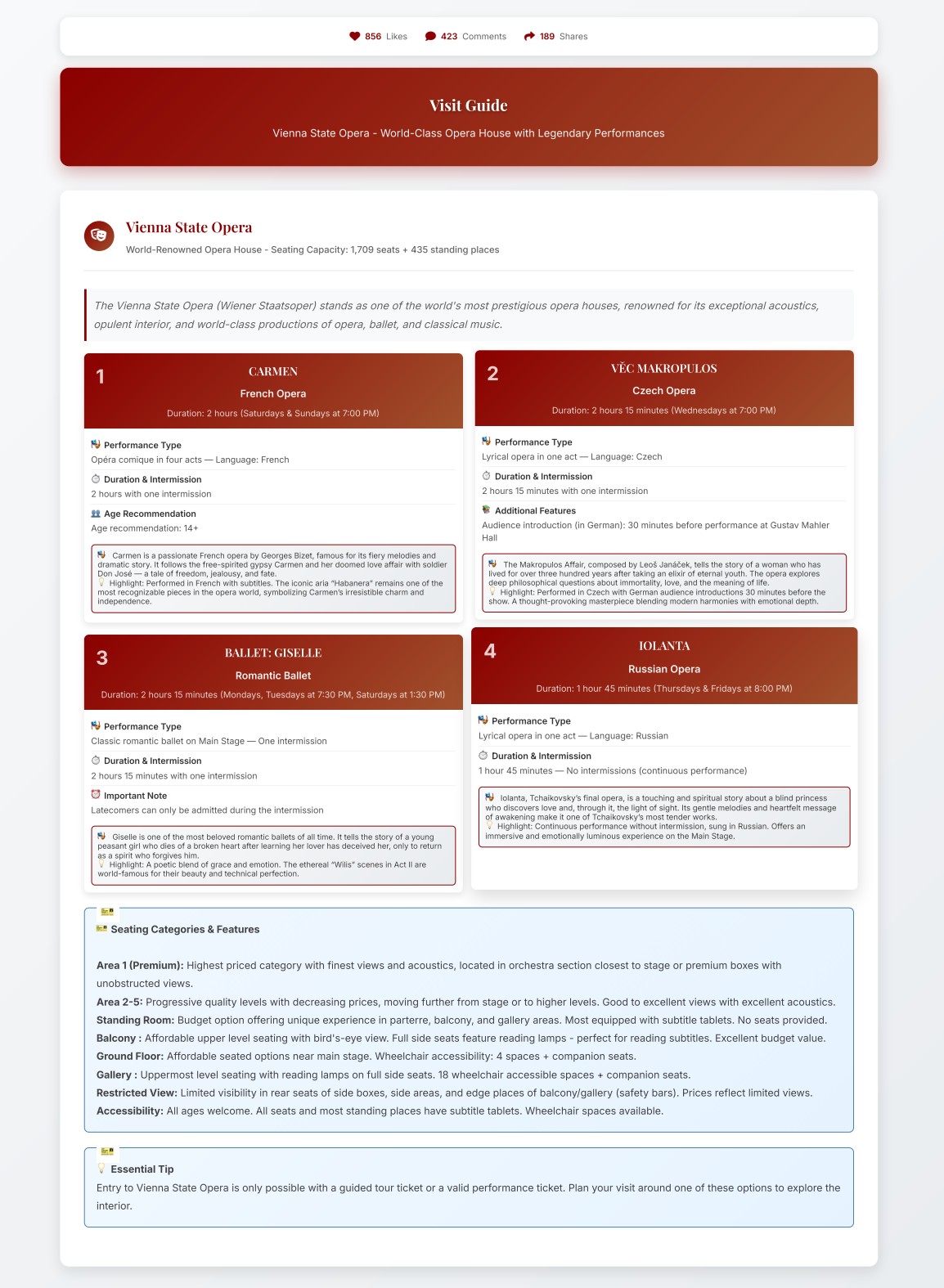

*Figure 12.* Attraction travel guide presenting visit recommendations in a blog-style format.

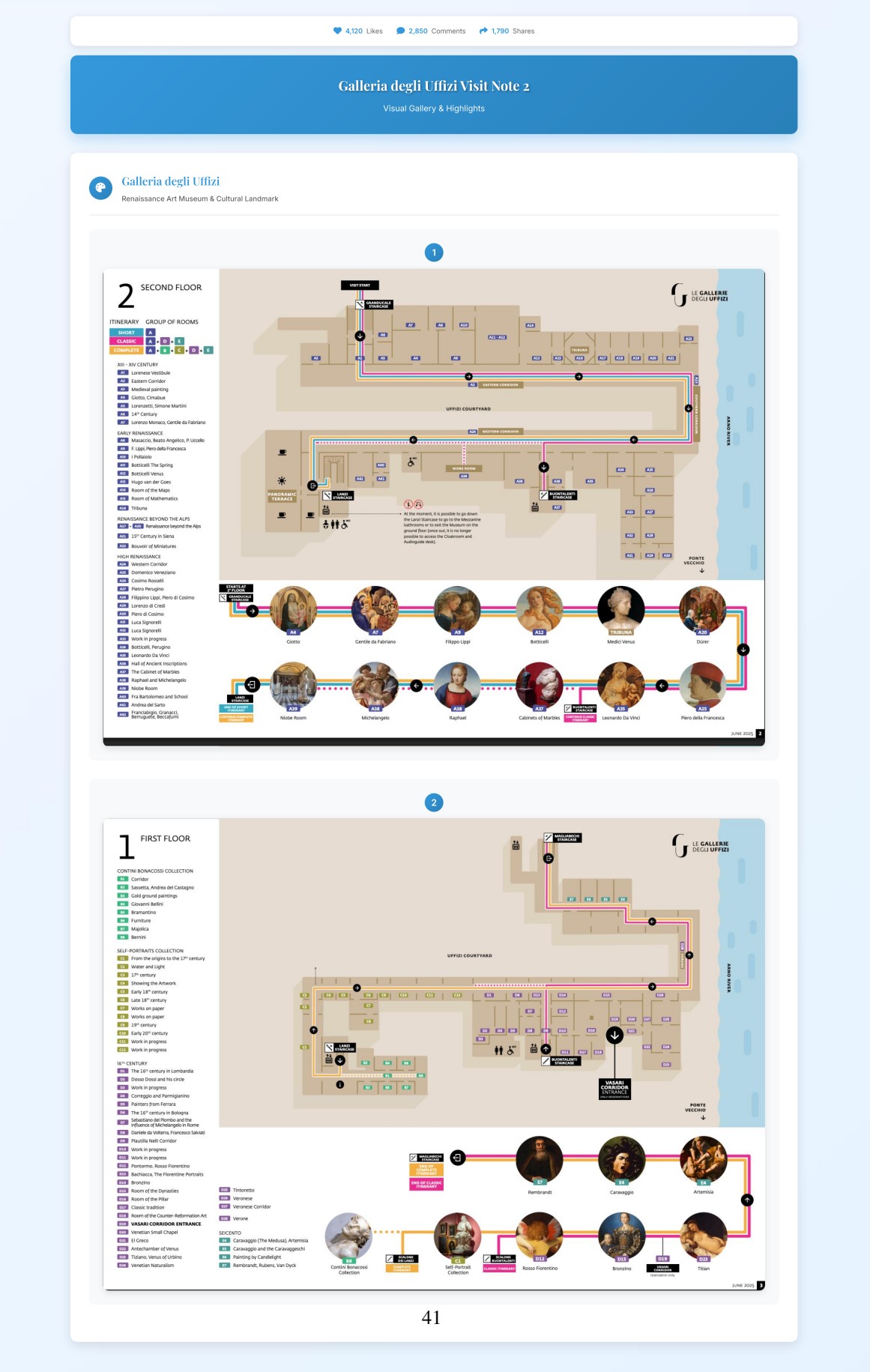

*Figure 13.* Scenic spot floor plan showing spatial layout and internal structure of an attraction.

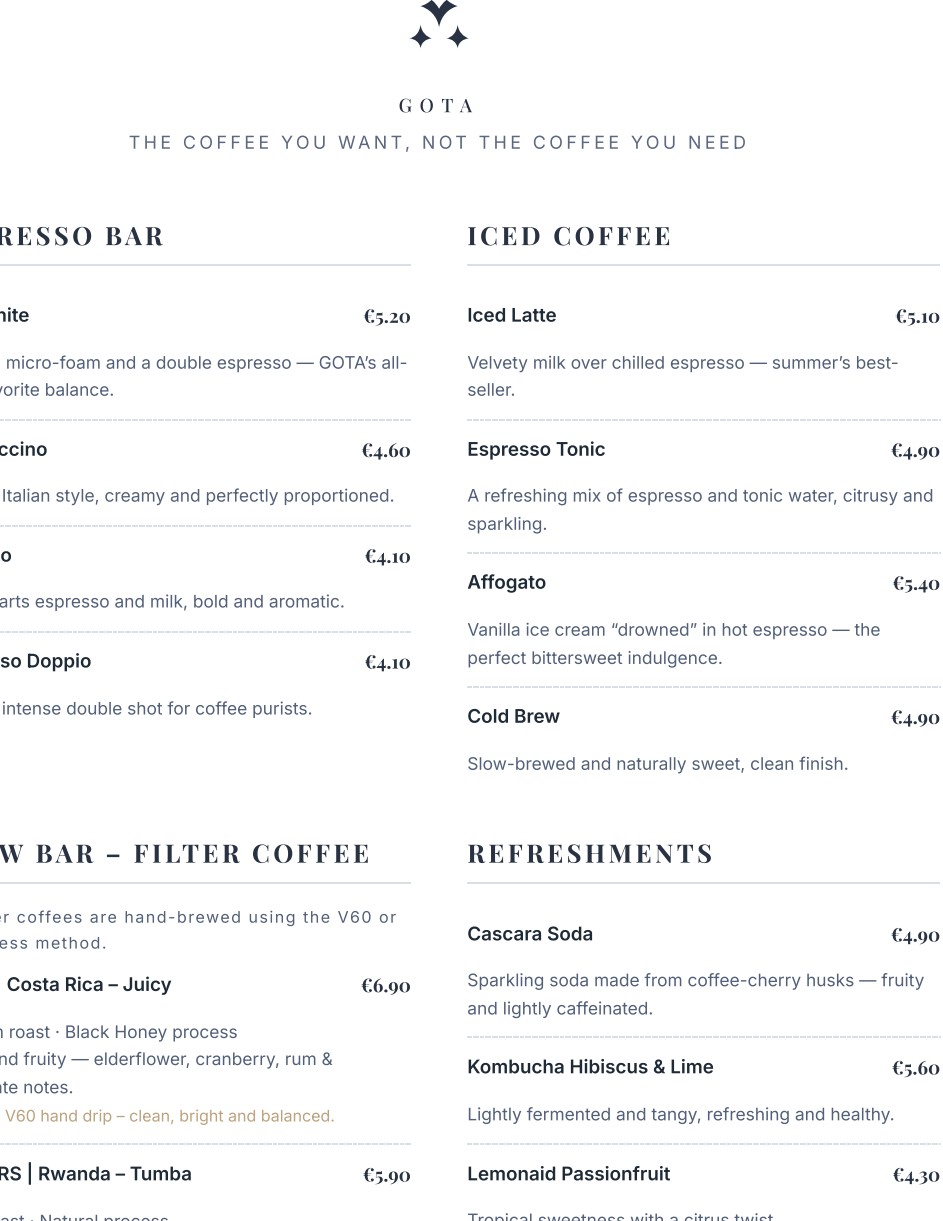

**GOTA**

THE COFFEE YOU WANT, NOT THE COFFEE YOU NEED

## ESPRESSO BAR

**Flat White**     €5.20

Smooth micro-foam and a double espresso — GOTA's all-time favorite balance.

**Cappuccino**     €4.60

Classic Italian style, creamy and perfectly proportioned.

**Cortado**     €4.10

Equal parts espresso and milk, bold and aromatic.

**Espresso Doppio**     €4.10

A pure, intense double shot for coffee purists.

## ICED COFFEE

**Iced Latte**     €5.10

Velvety milk over chilled espresso — summer's best-seller.

**Espresso Tonic**     €4.90

A refreshing mix of espresso and tonic water, citrusy and sparkling.

**Affogato**     €5.40

Vanilla ice cream "drowned" in hot espresso — the perfect bittersweet indulgence.

**Cold Brew**     €4.90

Slow-brewed and naturally sweet, clean finish.

## BREW BAR – FILTER COFFEE

All filter coffees are hand-brewed using the V60 or AeroPress method.

**GOTA | Costa Rica – Juicy**     €6.90

Medium roast · Black Honey process
Floral and fruity — elderflower, cranberry, rum & chocolate notes.
Method: V60 hand drip – clean, bright and balanced.

**FATHERS | Rwanda – Tumba**     €5.90

Light roast · Natural process
Sweet and rich — pineapple, milk chocolate & gingerbread finish.
Method: AeroPress – smooth, full-bodied and mellow.

## REFRESHMENTS

**Cascara Soda**     €4.90

Sparkling soda made from coffee-cherry husks — fruity and lightly caffeinated.

**Kombucha Hibiscus & Lime**     €5.60

Lightly fermented and tangy, refreshing and healthy.

**Lemonaid Passionfruit**     €4.30

Tropical sweetness with a citrus twist.

## COFFEE COCKTAILS

## PATISSERIE

*Figure 14.* Restaurant menu page with dish listings, prices, and time-dependent offerings.

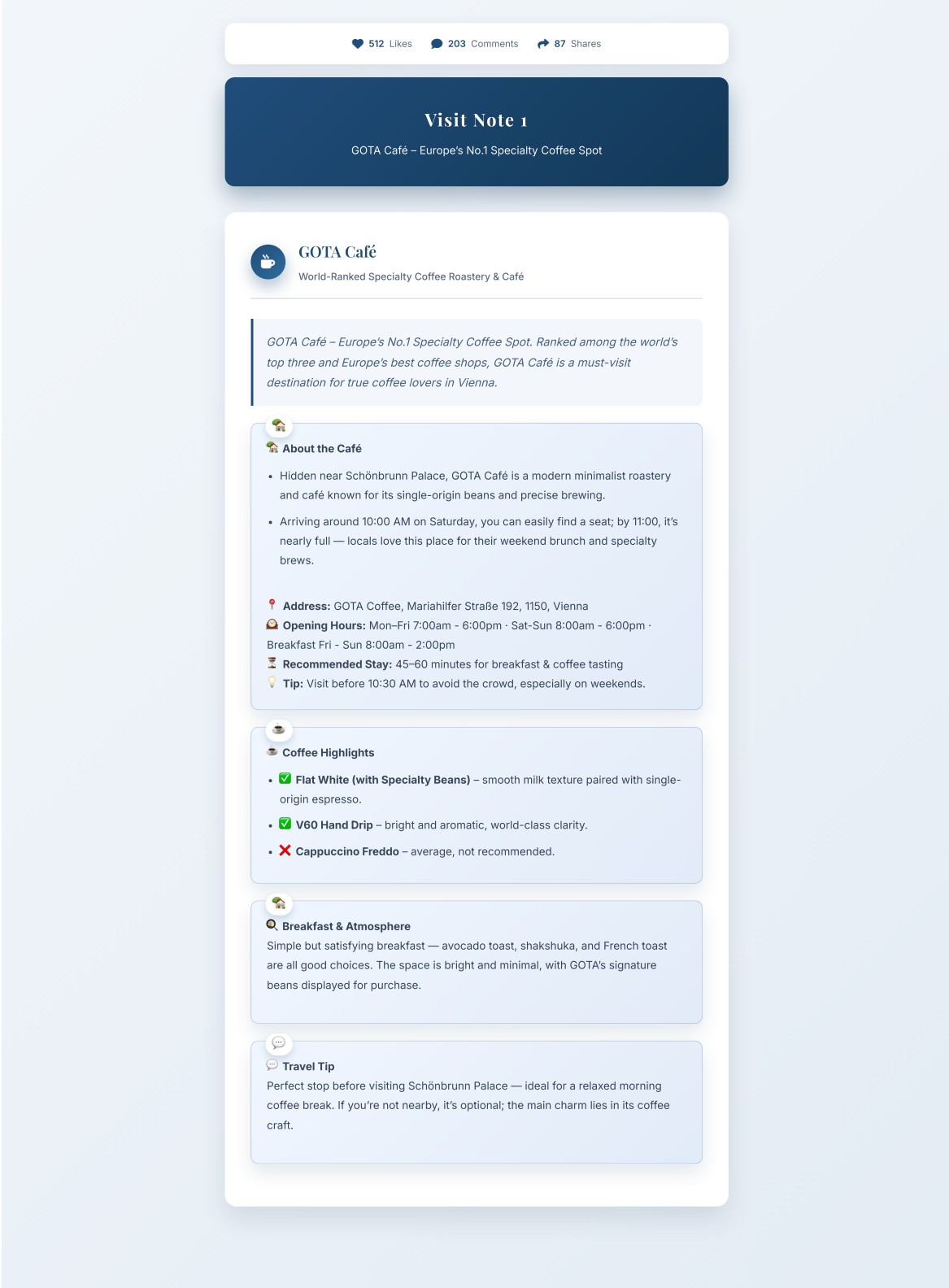

*Figure 15.* Restaurant travel notes combining subjective commentary with practical dining information.

