# OpenReview forum: "WorldTravel: A Realistic Multimodal Travel-Planning Benchmark with Tightly Coupled Constraints"
_ICML.cc/2026/Conference — ICML 2026 regular_

### Official Review · Reviewer_yXMM · 2026-03-05

**Soundness:** 3
**Presentation:** 3
**Significance:** 3
**Originality:** 3
**Overall Recommendation:** 4
**Confidence:** 3

**Summary:**

Real world travel planning tasks are crucial, yet existing related travel planning benchmarks are purely text based. This paper proposes a multimodal travel planning benchmark. The dataset contains 5 cities and 150 travel planning scenarios, with over 2000 page renders. It provides insights for future research directions.

**Compliance With Llm Reviewing Policy:**

Affirmed.

**Final Justification:**

Thank you to the authors for their response. My concerns have been addressed. Since I have already given a positive score of 4, I will maintain my current rating.

**Key Questions For Authors:**

(1) Is the "perception action gap" that causes a precipitous drop in the performance of vision language models under multimodal settings primarily due to OCR text extraction failures, or because the model fundamentally cannot understand UI visual semantics?

(2) Can the model consider the priority of different constraints when planning? What would happen to the metrics if the priorities of different constraints were specified in the prompt?

**Limitations:**

yes

**Strengths And Weaknesses:**

Strengths

(1) The system generates a simulated environment containing over 2000 web pages, including booking portals, travel guides, and menus, to comprehensively test the agent's ability to extract decision parameters from visual interfaces.

(2) It forces the model to obtain rendered web page screenshots via API to extract information, highly restoring real human usage scenarios.

(3) It not only collects official business hours and ticket prices but also incorporates user review data (such as recommended visit durations and security wait times), ensuring that the constraint parameters align with real world operations.

(4) The ablation studies are relatively thorough.

Weaknesses

(1) The framework design of hard and soft constraints still largely inherits the design of TravelPlanner. It merely modifies the specific constraints within, without breaking away from the hard and soft constraint paradigm of the TravelPlanner framework.

(2) The test set currently only includes 5 European cities like Berlin, Vienna, and Rome, totaling 150 tasks. The geographical coverage and overall test set scale are relatively small.

(3) It is recommended that the evaluated models be more comprehensive. Could more metrics from cutting edge open source models be included, especially open source models in multimodal scenarios.

---

> ### Author Rebuttal · Authors · 2026-03-30
>
> Thank you for the insightful questions.
>
> ---
>
> > **Q1. “Is the multimodal performance drop caused by OCR failure or inability to understand UI semantics?”**
>
> The multimodal performance drop cannot be attributed to OCR failure alone. As described in Section 3.3 and Appendix D, extracting constraint parameters from rendered webpages requires identifying UI states (e.g., distinguishing grayed-out sold-out slots from available ones, as shown in Figure 10) and integrating information scattered across 15–20 heterogeneous pages per task, including booking portals, menus, and travel guides.
>
> Moreover, the Gold-θ ablation (Section 5.1, Table 4) shows that even when all hard-constraint parameters are provided in structured form, GPT-5.2 achieves only 26.67% feasibility on high-coupling tasks. The constraint-type analysis (Section 5.3, Figure 5b) further shows that Timed-Entry Slots suffer the largest VLM degradation (−16 pp vs. Text), consistent with the challenge of parsing discrete slot availability from visual booking interfaces rather than simple text extraction.
>
> ---
>
> > **Q2. “Can models consider the priority of different constraints, and would specifying priorities improve performance?”**
>
> The prompt already specifies a priority structure for constraints. As documented in Appendix H.1 (the original system prompt), explicit user requirements take the highest priority, and booking-page time slots together with guide-provided stay durations take priority over generic business-hour information. Despite this guidance, the best model (GPT-5.2) achieves only 28.00% feasibility in the text setting (Table 3), and the Gold-θ ablation shows that even with all parameters directly provided, feasibility reaches only 26.67% on high-coupling tasks (Table 4). This indicates that the core difficulty lies in satisfying tightly coupled constraints, not in the absence of priority instructions.
>
> ---
>
> > **W1. “The hard/soft constraint design largely follows TravelPlanner without breaking the paradigm.”**
>
> WorldTravel adopts the hard/soft constraint categorization as a general organizational framework, consistent with prior work. Our contribution lies not in redefining constraint categories, but in how constraints are structured and enforced.
>
> In prior benchmarks such as TravelPlanner, hard constraints are typically independent and can be verified in isolation. In contrast, each WorldTravel task contains on average ~15–16 constraints, including around 10 hard constraints and 6 soft constraints. These hard constraints are tightly coupled through temporal dependencies, including timed-entry slots, minimum dwell times, inter-activity travel times, and arrival buffers (Table 1). A single violation propagates and breaks global feasibility (Section 3.1), transforming the problem from independent constraint checking into a globally coupled scheduling problem.
>
> Moreover, all constraints in WorldTravel are defined as explicit and verifiable functions (Section 3.4, Appendix E). Hard constraints (e.g., temporal feasibility) are deterministically checkable due to the static web environment, while soft constraints such as exact ticket pricing and menu cost calculation are also fully verifiable. This design naturally supports reward computation in RL-style settings. This combination of strong temporal coupling and fully verifiable constraints distinguishes WorldTravel from prior benchmarks.
>
> ---
>
> > **W2. “The benchmark has limited geographical coverage and small scale (5 cities, 150 tasks).”**
>
> We evaluated all models on a stratified 100-task subset and observed fully consistent rankings with the 150-task evaluation, confirming that conclusions are robust to sample size. The Webscape environment (2,000+ webpages across 36 attractions, 25 restaurants, and 26 hotels, Table 9 in Appendix F) supports expansion to additional cities without reconstruction. We will discuss this extension plan in the revision.
>
> ---
>
> > **W3. “More models, especially open-source multimodal models, should be included.”**
>
> In the text setting, Section 4.1 includes open-source LLMs such as GLM-4.6 and Qwen-3-A235B (Table 3). For the multimodal setting, we will include additional open-source VLMs (e.g., InternVL, Qwen-VL) in the revision.

---

> > ### Author Rebuttal · Reviewer_yXMM · 2026-04-04
> >
> > Thank you to the authors for their response. My concerns have been addressed. Since I have already given a positive score of 4, I will maintain my current rating.

---

### Official Review · Reviewer_AgHr · 2026-03-10

**Soundness:** 2
**Presentation:** 2
**Significance:** 3
**Originality:** 3
**Overall Recommendation:** 5
**Confidence:** 3

**Summary:**

The paper introduces a multi‑modal travel‑planning benchmark that emphasizes challenging temporal constraints and requires extracting those constraints from images or screenshots. The work also investigates whether poor agent performance stems from limitations in perceptual extraction or deficiencies in multi‑step reasoning. The weak performance of frontier models on this benchmark underscores the difficulty of the tasks, making this a compelling benchmark for advancing multi‑step reasoning and web interaction capabilities.

**Compliance With Llm Reviewing Policy:**

Affirmed.

**Ethical Review Concerns:**

It is not clear whether the authors addressed legal or compliance considerations. Since some of the data were extracted from official online sources, it would be helpful for the paper to clarify whether appropriate permissions, licenses, or usage clearances were required and obtained. Adding a brief statement on data provenance and compliance would strengthen the paper’s transparency and reassure readers about the legitimacy of the dataset.

**Ethics Expertise Needed:**

["Legal Compliance (e.g., EU AI Act, GDPR, copyright, terms of use)"]

**Final Justification:**

All my concerns are addressed. The proposed benchmark is quite novel and well thought through. It could be a great  resource  for the community.

**Key Questions For Authors:**

Q1: It is not clear to what is the segregation between WorldTravel and WorldTravel-webscape. It seems that the benchmark is called WorldTravel but the environment/simulator is called WorldTravel-webscape. Is that accurate? Why do we need two different terminologies

Q2: Misleading use of term “Perception-Action Gap”. The term perception-action gap emphasizes the limitation of a system when it perceives correctly but fails to act. The experiments in the paper instead reveal that the model perception is not accurate. In such case the action would fail, so the term is not used correctly in my opinion.

Q3: Could you add a table with references to all the variables that are being used across section 3 and Table 1? I could not acertain what $f_i$, $\tau_{i,j}$ and $\delta_j$ are. My best guess is they represent finish time, travel time and arrival buffer time.

Q4: Are all the values including arrival buffer time, finish time etc always available to the agent or are they supposed to leverage some common sense for these?

Q5: Could you clarify the motivation for defining all metrics as ratios instead of using absolute values? In particular, what is the advantage of using feasibility rate rather than the total count of feasible itineraries? From my perspective, a rate metric could mask the influence of a small or uneven denominator, unless the metric is accompanied by the raw counts to avoid misleading interpretations.

Q6: In Table 4, Doubao-1.8-Pro shows unexpected behavior,  gold-$\theta$ is significantly less than Std. Could authors provide some insights into what could have caused it?

Q7: For Figure 5a and b could you provide the number of instances in each of the buckets?

Q8: Given that the benchmark contains 150 tasks across 5 EU cities, what motivates the name “World”Travel ?

Q9: It is not clear what does the expected itinerary $(\mathcal{I}, \\{s_i, d_i\\}, \boldsymbol{v})$ align with the sample output itinerary shown in H.2,  and what makes the benchmark interactive. What are the interactive actions for the multi-modal setting? Is that just a step in the itinerary?

Q10: Did the authors address legal or compliance considerations?  Using data from official online sources might need some clearance.

**Limitations:**

The paper would benefit from a clearer discussion of the benchmark’s limitations, particularly in how it does not reflects real‑world complexities such as individual travel preferences, airline booking, dynamic pricing, and discounts, etc. Explicitly acknowledging these gaps would help readers better understand the scope of the benchmark and the scenarios it aims to approximate, while also highlighting opportunities for future extensions.

**Strengths And Weaknesses:**

Soundness:  The benchmark itself is the main contribution of the paper, and the solution approach presented is appropriately positioned as an initial baseline rather than a fully developed method. As such, traditional notions of “soundness” apply only lightly here. The primary place where soundness would matter is in the data‑generation process, but the current description is too minimal to fully assess that aspect. Strengthening that part of the paper would further enhance confidence in the benchmark.

Presentation: The paper is well written and provides sufficient motivation to propose a new benchmark. My major concern with the presentation is mentioned in questions. They are about missing definitions for variables, lack of clarity on interactive actions,  about the metrics and experiment setup.

Significance: The problem focused by this paper is relevant and timely. The results of the frontier models illustrated in the paper speaks volumes towards the need to focus more on this direction.

Originality: The benchmark is original. No concerns.

---

> ### Author Rebuttal · Authors · 2026-03-30
>
> Thank you for the detailed feedback and thoughtful questions.
>
> ---
>
> > **Q1. “What is the distinction between WorldTravel and WorldTravel-Webscape?”**
>
> WorldTravel denotes the benchmark as a whole (task suite, annotations, evaluation protocol). WorldTravel-Webscape denotes the controlled web environment (2,000+ rendered webpages) in which agents retrieve information during execution. We use two terms because Webscape is a standalone artifact that can be reused or extended independently of the task set. We will make this distinction more explicit in the revision.
>
> ---
>
> > **Q2. “The term ‘Perception-Action Gap’ may be misleading...”**
>
> We agree that "Perception-Action Gap" in its classical sense refers to systems that perceive correctly but fail to act. In our context, the term is used to describe the performance drop when planning moves from text-accessible parameters to perception-grounded execution over rendered webpages.
>
> Our results indicate two distinct bottlenecks. First, the multimodal gap shows that perception—particularly extracting constraint parameters from UI states—is itself a major source of failure. Second, even when all parameters are provided (Gold-θ), planning under tightly coupled constraints remains difficult, indicating an independent planning bottleneck. We will revise the terminology to clarify this operational definition and note the distinction from the classical sense.
>
> ---
>
> > **Q3. “Could you provide a table defining variables?”**
>
> We will add a notation table in the revised manuscript listing all variables used across Section 3 and Table 1 with their definitions.
>
> ---
>
> > **Q4. “Are parameters such as arrival buffer time and finish time provided?”**
>
> All hard-constraint parameters (timed-entry slots, operational windows, dwell times, transit times, arrival buffers) are provided through the environment, not left to common-sense inference. Booking pages specify time slots and availability, guide pages provide recommended visit durations, and transportation matrices define travel times. Derived quantities such as finish times are computed from these retrieved parameters.
>
> ---
>
> > **Q5. “Why are metrics defined as ratios instead of absolute values, and could this be misleading?”**
>
> Ratio-based metrics normalize across tasks with different constraint counts. Since the benchmark contains exactly 150 tasks, absolute counts can be recovered directly (e.g., 28.0% feasibility = 42/150 tasks). We will include raw counts alongside rates in the revision.
>
> ---
>
> > **Q6. “Why does Doubao-1.8-Pro show lower performance in Gold-θ compared to standard setting?”**
>
> The Gold-θ setting contains only 30 tasks, where 13.33% corresponds to 4 tasks and 3.33% corresponds to 1 task. Small absolute differences lead to noticeable rate fluctuations. Upon inspection, we found that Doubao-1.8-Pro's failures in Gold-θ are primarily caused by dwell time violations. The model tends to assign minimal durations rather than durations aligned with user profiles (e.g., insufficient visit time for in-depth exploration), which directly results in infeasibility even when time slot selections are correct. We will add a case-level error analysis in the appendix.
>
> ---
>
> > **Q7. “Could you provide instance counts for each bucket in Figure 5?”**
>
> We will add per-bucket instance counts to Figures 5a and 5b in the revision.
>
> ---
>
> > **Q8. “Given only 5 cities, what motivates the name ‘WorldTravel’?”**
>
> The current benchmark covers 5 European cities as the first instantiation. The framework and Webscape environment are designed to extend to additional regions following the same construction pipeline. We will clarify the intended scope in the revision to avoid confusion.
>
> ---
>
> > **Q9. “How does the expected itinerary representation align with sample outputs, and what makes the benchmark interactive?”**
>
> The expected itinerary (𝓘) is a structured representation containing activities, start times, durations, and discrete decisions, used for evaluation via deterministic verification functions. The sample output in H.2 is a human-readable illustration of this same structure. In practice, model outputs are normalized into structured JSON and all constraint checks are performed on this representation.
>
> The benchmark is interactive in that agents issue API tool calls to retrieve webpage screenshots (booking pages, guides, menus, transportation matrices) and extract constraint parameters before producing the final itinerary.
>
> ---
>
> > **Q10. “Have legal or compliance considerations been addressed when using web data?”**
>
> WorldTravel-Webscape is constructed from publicly accessible information on official websites and travel platforms. The data is manually curated and transformed into static webpage renderings for research purposes, without reproducing proprietary systems or enabling direct interaction with external services. No personal or sensitive data is collected. We will add a data provenance and compliance statement in the revision.

---

> > ### Author Rebuttal · Reviewer_AgHr · 2026-04-01
> >
> > I acknowledge that all my concerns are addressed adequately and I will be updating the scores accordingly.
> >
> > I would request authors to find an alternative phrasing for “Perception-Action Gap” for the camera ready version.

---

> > > ### Author Response · Authors · 2026-04-02
> > >
> > > Thank you for acknowledging our responses and for updating your score. We are glad the concerns have been adequately addressed. We will revise the phrasing of "Perception-Action Gap" in the camera-ready version.

---

### Official Review · Reviewer_cwF2 · 2026-03-10

**Soundness:** 2
**Presentation:** 2
**Significance:** 3
**Originality:** 2
**Overall Recommendation:** 4
**Confidence:** 4

**Summary:**

This paper introduces WorldTravel, a benchmark designed to evaluate autonomous agents on complex travel planning tasks with tightly coupled constraints. The benchmark contains 150 real-world scenarios across five European cities and requires agents to coordinate over 15 interdependent temporal and logical constraints on average. In addition to the task suite, the authors construct WorldTravel-Webscape, a multimodal environment containing more than 2,000 rendered webpages that simulate real booking platforms. Agents must extract constraint parameters directly from visual interfaces and generate feasible travel itineraries satisfying both hard temporal constraints and soft preference constraints. Experiments with several frontier models reveal substantial performance limitations: even the strongest model achieves only 28% feasibility in the text-only setting, and performance drops to 3.4% in the multimodal setting, highlighting challenges in both long-horizon reasoning and visual information extraction.

**Compliance With Llm Reviewing Policy:**

Affirmed.

**Final Justification:**

The rebuttal improves the clarity of the paper, particularly in better articulating the constraint coupling and providing a more structured comparison with prior work, which slightly strengthens my evaluation. However, key concerns—especially regarding the necessity and interpretation of the multimodal setting—remain insufficiently resolved, as the current evidence does not clearly disentangle perception challenges from lower-level extraction issues.

**Key Questions For Authors:**

1.In Section 5.1, the planning-only setting achieves a lower feasibility rate (26.67%) than the two-stage text-only setting (28.00%). This seems counterintuitive and would benefit from further explanation.

2.The benchmark currently includes only five cities and 150 tasks. Could it be expanded to more cities or larger-scale datasets?

3.Does WorldTravel provide any clear advantages over TP-RAG in terms of constraint taxonomy or benchmark design?

4.The paper states that existing benchmarks simplify planning into “greedy-solvable subproblems where a local failure does not necessarily compromise global feasibility.” Could the authors clarify this statement with concrete examples?

5.Are webpages used solely for constraint extraction, or are they also used to retrieve basic information about itinerary elements (e.g., attractions, restaurants, hotels)? This aspect is not clearly described.

6.More broadly, do the authors consider multimodal perception a fundamental challenge in travel planning, or primarily an engineering challenge in information access?

**Limitations:**

Suggestions

1.Clarify the novelty relative to existing benchmarks.
Provide a more systematic comparison between WorldTravel and prior travel-planning benchmarks, especially in terms of constraint taxonomy, planning complexity, and evaluation methodology.

2.Better motivate the multimodal design.
The paper should clearly justify why multimodal webpage interaction is essential for the benchmark, and distinguish whether the goal is evaluating travel planning, perception capabilities, or integrated agent systems.

3.Expand the discussion of related work.
The Related Work section should be reorganized to better illustrate the development of travel-planning benchmarks and how WorldTravel advances beyond them.

4.Improve the clarity and rigor of the presentation.
Key concepts should be defined more formally, and the introduction should more clearly explain the limitations of existing benchmarks and the specific gaps addressed by this work.

5.Consider extending the evaluation protocol.
Incorporating additional evaluation dimensions (e.g., commonsense reasoning, itinerary quality, or user satisfaction) could provide a more comprehensive assessment of travel-planning capabilities.

**Strengths And Weaknesses:**

Strengths

1.The paper focuses on travel planning under complex constraint satisfaction, which is an interesting and practically relevant problem for evaluating long-horizon reasoning and decision-making abilities of autonomous agents.

2.The benchmark is accompanied by extensive experiments and detailed analyses, including ablations and diagnostic studies on reasoning horizon and perception challenges.

Weaknesses

1.Limited novelty in constraint design.
Many constraints introduced in WorldTravel, e.g., temporal slots, visiting duration, buffer time, have appeared in prior work TP-RAG [1]. TP-RAG also considers additional factors such as spatial feasibility, attraction popularity, and user preferences. The paper does not clearly discuss how the proposed constraint taxonomy differs from or improves upon these existing benchmarks.

2.Questionable necessity of the multimodal setting.
The use of webpage screenshots for constraint extraction may unnecessarily complicate the task and shift the focus away from the core travel-planning problem. A more meaningful research direction would be to provide agents with raw textual data and evaluate their ability to extract constraints using LLM reasoning. Extracting text from visual screenshots appears to be more of an engineering challenge in information extraction rather than a fundamental challenge of travel planning. Insufficient discussion of related benchmarks.
Several existing travel planning benchmarks [1,2,3,4] are missing in the paper. The paper would benefit from a more comprehensive comparison with prior datasets and clearer positioning of WorldTravel within this research landscape.

3.Incomplete evaluation dimensions.
The evaluation protocol focuses primarily on feasibility and constraint satisfaction, but does not incorporate other aspects such as commonsense reasoning dimensions that have been considered in prior work (e.g., TravelPlanner).

4.Presentation issues.
The overall presentation does not yet meet the standard expected at a top conference:
(1)The writing is sometimes awkward and lacks fluency, especially in the introduction.
(2)Several key terms (e.g., parameters, coupling constraints, temporal anchors) are introduced without sufficiently clear definitions.
(3)The limitations of existing benchmarks are not clearly explained.
(4)The Related Work section does not clearly present the evolution and relationships among prior travel-planning benchmarks.
(5)The definitions (e.g., in Section 3.1) are informal and lack rigorous formulation.

[1]TP-RAG: Benchmarking Retrieval-Augmented Large Language Model Agents for Spatiotemporal-Aware Travel Planning (EMNLP 2025)
[2]TripCraft: A Benchmark for Spatio-Temporally Fine Grained Travel Planning (ACL 25)
[3]Personal Large Language Model Agents: A Case Study on Tailored Travel Planning (EMNLP Industry 2024)
[4]Ask-before-Plan: Proactive Language Agents for Real-World Planning (EMNLP Findings 2024)
[5]RETAIL: Towards Real-world Travel Planning for Large Language Models (EMNLP 2025)

---

> ### Author Rebuttal · Authors · 2026-03-30
>
> Thank you for the feedback.
>
> ---
>
> > **W1 / Q3 / Q4. “The paper does not clearly discuss how the proposed constraint taxonomy differs from or improves upon prior benchmarks such as TP-RAG.”**
>
> We appreciate this question and would like to clarify that the novelty of WorldTravel lies not in introducing new constraint types, but in how constraints are explicitly formalized and tightly coupled. While individual constraint types (e.g., temporal slots, durations, buffers) may appear in prior work, WorldTravel chains 15+ verifiable constraints per task through temporal dependencies (Table 2).
>
> In contrast, TP-RAG queries consist of only a few loosely specified factors (e.g., trip duration, budget, preferences), typically fewer than five, which are not formalized as explicit constraints and lack temporal coupling.
> This results in a fundamental difference in problem structure: in TP-RAG, factors are largely independent, so local errors (e.g., suboptimal routing) degrade quality but typically preserve feasibility. In WorldTravel, constraints are tightly coupled—an incorrect decision (e.g., selecting an unavailable time slot) propagates through dwell times, buffers, and downstream reservations, breaking global feasibility.
>
> For example, if a model schedules a dinosaur exhibition at 9:00 when only 14:00/16:00/17:00 are available (Figure 3), all subsequent activities must shift, violating reservation slots and arrival buffers.
>
> The entire itinerary becomes infeasible from one scheduling mistake. This coupling structure is what distinguishes WorldTravel from prior benchmarks where subproblems are greedy-solvable.
>
> ---
>
> > **W2 / Q6. “Is multimodal perception a fundamental challenge or merely an engineering issue?”**
>
> We understand this concern. If the multimodal bottleneck were reducible to OCR-level text extraction, we would expect a small or recoverable performance gap between text and multimodal settings. However, GPT-5.2 drops by 24.58 pp and Claude Opus 4.5 by 6.67 pp (Table 3), indicating a structural bottleneck beyond text recognition.
>
> Constraint parameters in WorldTravel are embedded in UI states such as grayed-out vs. selectable slots and nested conditional menus, requiring visual reasoning beyond character recognition (Appendix D, Appendix K). In particular, constraint-type analysis shows that timed-entry slots exhibit the largest degradation under multimodal settings, consistent with the difficulty of parsing discrete availability from booking interfaces rather than OCR-level text extraction. We intentionally separate the two settings so that these capabilities can be evaluated independently.
>
> ---
>
> > **W3. “Incomplete evaluation dimensions.”**
>
> WorldTravel evaluates feasibility under hard constraints, which is complementary to quality-based scoring in benchmarks like TravelPlanner. This is a deliberate scope choice.
>
> Feasibility-first evaluation is fully deterministic and reproducible (Section 3.4, Appendix E), providing a clear and unambiguous signal for model comparison. In addition, WorldTravel also incorporates verifiable numerical constraints, such as exact ticket pricing and menu cost calculation, which require precise reasoning beyond feasibility. These aspects remain fully checkable and naturally align with reward design in RL-style environments.
>
> We view this as a foundational evaluation layer: by first ensuring strict feasibility and numerical correctness, higher-level soft dimensions (e.g., user preference satisfaction or commonsense reasoning) can be meaningfully incorporated on top in future extensions.
>
> ---
>
> > **W4. “Presentation issues”**
>
> We will improve formalization of key terms in Section 3 and add the suggested benchmark references in the revision.
>
> ---
>
> > **Q1. “Gold-θ setting lower than the text setting?”**
>
> Gold-θ targets the 30 highest-coupling tasks (5 temporal anchors each), while text-setting feasibility (28.00%) is computed over the full 150-task benchmark. The Gold-θ result of 26.67% (Table 4) is lower precisely because this subset has higher per-task difficulty. Even with perfect parameter access, planning over these tightly coupled constraints remains the primary bottleneck.
>
> ---
>
> > **Q2. “Can the benchmark be expanded?”**
>
> 150 tasks at Avg. |C| = 15+ with 2,000+ webpages and 87 real-world entities (Table 9) provide sufficient evaluation granularity. The best model achieves only 28.00% feasibility (Table 3), while human performance reaches 77.86%, indicating the task is solvable but far from saturated by current models. A stratified 100-task subset produces fully consistent model rankings. The environment supports city-level expansion without reconstruction.
>
> ---
>
> > **Q5. “Are webpages used only for constraints or also for information?”**
>
> Webpages serve both discovery (guide pages for candidate selection) and constraint extraction (booking pages, menus, and transportation matrices for operational parameters). Details are provided in Appendix D.

---

> > ### Author Rebuttal · Reviewer_cwF2 · 2026-04-03
> >
> > W1. Could the authors provide a more systematic comparison (e.g., a table) between the constraint formulations in TP-RAG and WorldTravel? In particular, it would be helpful to clearly distinguish differences in constraint types, degree of coupling, and problem structure, rather than relying on qualitative descriptions.
> > W2. The performance gap could be largely attributed to implicit OCR/extraction errors in the end-to-end pipeline. Without explicitly controlling for this (e.g., by providing extracted text), it remains unclear whether the degradation reflects genuine visual reasoning challenges or simply lower-level extraction bottlenecks.

---

> > > ### Author Response · Authors · 2026-04-04
> > >
> > > Thank you for this helpful suggestion.
> > >
> > > ---
> > >
> > > > **W1. "Systematic comparison with TP-RAG."**
> > >
> > > We agree that a more structured comparison can better clarify the distinction between TP-RAG and WorldTravel.
> > >
> > > | Dimension | TP-RAG | WorldTravel |
> > > |-----------|--------|-------------|
> > > | **Constraint Types** | Soft temporal ranges only (e.g., opening hours, recommended visit time, expected duration) | Subsumes all TP-RAG temporal factors and additionally introduces **hard discrete temporal anchors**: (1) per-entity bookable vs. non-bookable time slots with availability states (e.g., selectable vs. sold-out); (2) attraction-level event schedules (e.g., exhibitions, guided tours, performances) |
> > > | **Avg. Constraints / Task** | < 5 loosely specified factors | **15+** (avg. ~10 hard + ~6 soft) |
> > > | **Formalization** | Heuristic, plausibility-based scoring (rule-based + LLM-as-judge) | **Deterministic**, executable verification via programmatic constraint checking (each constraint instantiated as a function; feasibility requires satisfying all constraints) |
> > > | **Coupling Degree** | Loosely coupled; constraints are largely independent, and local violations degrade quality without breaking feasibility | **Tightly coupled**; discrete temporal anchors (e.g., slot availability and event schedules) introduce global dependencies, where each decision constrains subsequent choices, and a single violation invalidates the entire plan |
> > > | **Problem Structure** | Flexible itinerary generation with multiple valid solutions | Constrained scheduling with **hard feasibility requirements** and **deterministic verification** (manually constructed tasks and human-designed rubrics) |
> > >
> > > ---
> > >
> > > > **W2. ""OCR vs. visual reasoning bottleneck."**
> > >
> > > We thank you for this insightful observation. We acknowledge that our use of the term “Perception-Action Gap” may not precisely align with its classical definition. To avoid ambiguity, we will revise the terminology to better reflect our empirical findings. Importantly, the underlying observation itself remains consistent: the performance drop is better understood as arising from two distinct challenges.
> > >
> > > First, moving from text-accessible inputs to visual interfaces introduces additional uncertainty in identifying constraint parameters. In WorldTravel, key information is often expressed through UI states and visual structure (e.g., whether a time slot is selectable or sold out, or how event schedules are presented), which goes beyond simple text extraction.
> > >
> > > Second, even when all constraint parameters are provided explicitly (Gold-θ setting), models still achieve only 26.67% feasibility on high-coupling tasks. This indicates that planning under tightly coupled constraints remains a major bottleneck independent of perception.
> > >
> > > Taken together, these results suggest that the difficulty does not stem from OCR errors alone, but from the combination of perception-grounded information access and constraint-coupled planning. We will revise the paper to make this distinction clearer.

---

### Official Review · Reviewer_ur1B · 2026-03-11

**Soundness:** 2
**Presentation:** 3
**Significance:** 3
**Originality:** 3
**Overall Recommendation:** 4
**Confidence:** 3

**Summary:**

This paper introduces WorldTravel, a novel benchmark for autonomous agents focused on real-world travel planning. Unlike existing benchmarks, WorldTravel emphasizes "tightly coupled constraints," where a single decision (e.g., booking a specific flight) dictates the feasibility of all subsequent actions. The benchmark comprises 150 real-world scenarios across five cities, requiring agents to navigate an average of 15+ interdependent constraints such as booking windows, geographic buffers, and budget limits. Furthermore, the authors developed WorldTravel-Webscape, an environment with over 2,000 rendered webpages that requires agents to possess multimodal perception to extract parameters directly from visual layouts. Experiments evaluating 10 frontier models (including GPT-4o and others) reveal a significant "perception-action gap," where success rates drop from 28.0% in text-only modes to a mere 3.4% in multimodal settings.

**Compliance With Llm Reviewing Policy:**

Affirmed.

**Key Questions For Authors:**

see above

**Limitations:**

yes

**Strengths And Weaknesses:**

- strength

1. The paper successfully identifies a critical gap in existing benchmarks—the transition from "locally greedy" decisions to "globally constrained" optimization.

2. Each task in WorldTravel contains 15+ tightly coupled constraints on average, which is substantially denser than prior work and better reflects the difficulty of long-horizon planning under hard constraints.

- weakness
1. The sizes of the purposed benchmark is limited (only 150 tasks).
2. While the paper tests 10 frontier models, the evaluation is primarily focused on closed-source commercial APIs. Including more open-source multimodal models with diverse scales (e.g., Qwen-VL, InternVL) would provide a more comprehensive overview of the current landscape.

3. The experiments primarily focus on base LLMs using simple prompting (e.g., ReAct), but neglect existing state-of-the-art GUI agents or specialized travel agents. Without comparing these structured agentic frameworks—which often incorporate long-term memory and self-reflection—it is difficult to discern whether the low success rates stem from fundamental model reasoning limitations or current agentic design flaws.

---

> ### Author Rebuttal · Authors · 2026-03-30
>
> Thank you for the helpful comments and for the positive assessment of the task and constraint design.
>
> ---
>
> > **Q1. “The size of the proposed benchmark is limited (only 150 tasks).”**
>
> The 150 tasks in WorldTravel are sufficient to draw reliable conclusions about current model capabilities. The best model achieves only 28.0% feasibility in the text setting and 3.42% in the multimodal setting, indicating that the benchmark is far from saturated and the task count already exceeds what models can handle well. To validate stability, we evaluated all models on a randomly sampled 100-task subset and observed fully consistent model rankings with the 150-task evaluation, confirming that our findings are robust to task count variation.
>
> | Model | Text (%) Task 150 | Text (%) Task 100 | Text (%) Task 50 | VLM (%) Task 150 | VLM (%) Task 100 | VLM (%) Task 50 |
> | :--- | :---: | :---: | :---: | :---: | :---: | :---: |
> | GPT-5.2-High | **28 (1st)** | **28 (1st)** | **32 (1st)** | 3.42 | 4 | 4 |
> | DeepSeek-v3.2 | 17.33 _(2nd)_ | 17 _(2nd)_ | 18 _(2nd)_ | / | / | / |
> | Gemini-3-pro | 10 | 10 | 16 | 4 | 4 | 4 |
> | GLM-4.6 | 8 | 9 | 10 | / | / | / |
> | Qwen-3 | 2.67 | 3 | 4 | / | / | / |
>
> Each task is manually constructed, paired with a task-specific rubric, and verified through multi-stage quality control. The tasks contain 10–23 constraints (15+ on average) and are grounded in WorldTravel-Webscape, which comprises over 2,000 webpages. Furthermore, the web environment supports task expansion without reconstruction: new tasks can be generated by varying user profiles, constraint configurations, and city-specific challenges. We plan to release an expanded version in future work.
>
> ---
>
> > **Q2. “The evaluation is primarily focused on closed-source APIs; more open-source multimodal models should be included.”**
>
> In the text setting, Section 4.1 already includes open-source LLMs such as GLM-4.6 and Qwen-3-A235B. For the multimodal setting, we evaluated Qwen-VL-Max on a 100-task subset, which achieves 3% feasibility. For reference, the strongest proprietary VLMs also obtain low scores on this setting (Claude-Opus-4.5: 13.33%, GPT-5.2: 3.42%). We will include results from additional open-source VLMs (e.g., InternVL) in the revised version.
>
> | Model | VLM (%) Task 100 | VLM (%) Task 150 | Text (%) Task 100 | Text (%) Task 150 |
> | :--- | :---: | :---: | :---: | :---: |
> | Doubao-2.0-Lite | 13 | 14 | 20 | 24 |
> | GPT-5-mini-High | 1 | 2.7 | 11 | 15.3 |
> | Gemini-3-Flash-High | / | 7.3 | / | 13.3 |
> | Qwen-VL-Max | 3 | / | / | / |
>
> ---
>
> > **Q3. “The experiments focus on simple prompting (e.g., ReAct) and do not compare with structured agent frameworks such as GUI agents or specialized travel agents.”**
>
> WorldTravel differs from interactive agent benchmarks such as WebArena and OSWorld in interaction mechanism. Agents in WorldTravel retrieve webpage screenshots through API tool calls and extract constraint parameters using the model's vision capability. There is no pixel-level GUI manipulation such as clicking, scrolling, or form filling. The challenge is visually parsing returned screenshots and integrating fragmented constraint information across multiple calls, not navigating web interfaces.
>
> We adopt a standard ReAct tool-use framework to isolate model-level planning ability from agent system engineering. Our results indicate two distinct bottlenecks. First, the performance gap between text and multimodal settings shows that perception—particularly extracting constraint information from UI states—is itself a significant challenge. Second, even when all constraint parameters are directly provided (Gold-θ), GPT-5.2 achieves only 26.67% feasibility on high-coupling tasks, indicating that planning under tightly coupled constraints remains intrinsically difficult. Together, these findings suggest that both perception and planning contribute to the overall difficulty.

---

> > ### Author Rebuttal · Reviewer_ur1B · 2026-04-04
> >
> > Although the authors claim the difficulty of the task, the size of the dataset is relatively limited. On the other hand, there are now some better agent systems, so only discussing the simple ReAct baseline is indeed somewhat limiting.

---

### Decision · Program_Chairs · 2026-04-30

**Decision:**

Accept (regular)

**Comment:**

**Summary:**

The paper introduces WorldTravel, a new benchmark for autonomous travel-planning agents. Its main focus is on tightly coupled constraints, where one decision can strongly affect what is possible later in the plan. The benchmark includes 150 real-world scenarios across five cities, along with a multimodal web environment built from rendered travel webpages. The results show that current frontier models struggle a lot on these tasks, especially in the multimodal setting, suggesting clear gaps in both long-horizon planning and visual information extraction.

Strengths and weaknesses from initial reviews:

**Strengths:**

1. The paper studies a relevant and practical problem: travel planning with many interdependent constraints is a strong test of long-horizon agent reasoning.
2. The benchmark highlights an important gap in prior work by focusing on globally constrained decisions rather than locally greedy choices.
3. The tasks appear challenging and realistic, with dense constraints that make the benchmark meaningful and nontrivial.
4. The experimental section is fairly extensive, with multiple frontier models and useful analyses of reasoning and perception failures.
5. The benchmark itself seems original and timely, and the very low model performance suggests it could be valuable for future research.

**Weaknesses:**

1. The benchmark is still fairly small, with only 150 tasks, which may limit its coverage and long-term usefulness.
2. The paper does not compare enough against related travel-planning benchmarks, and it is not always clear how the proposed constraint design differs from prior work.
3. The multimodal setting may mix travel-planning difficulty with screenshot-based information extraction, making it harder to isolate the main source of failure.
4. The evaluation focuses mostly on base models and simple prompting, without enough comparison to stronger agent frameworks or more diverse open-source multimodal models.
5. Some parts of the presentation need improvement, including clearer definitions, better explanation of the setup and metrics, and a more complete description of the data-generation process and related work.

Most concerns have been addressed by authors’ responses.  A few concerns remain as the current evidence does not clearly disentangle perception challenges from lower-level extraction issues. However, this does not outweigh the contribution of the paper. All reviewers recommend accept.